# Evaluation of methods for the measurement of antibody-dependent enhancement of dengue virus infection using different FcγRIIa expressing cell lines

Shweta Chelluboina, Darshan Kshirsagar, Gauri Panzade, Akhilesh Chandra Mishra, Vidya Arankalle, Shubham Shrivastava [ORCID]*

Translational Virology, Interactive Research School for Health Affairs (IRSHA), Bharati Vidyapeeth (Deemed to be University), Pune, India

* shubham.shrivastava@bharatividyapeeth.edu, shubhamniv@gmail.com

## Abstract

### Background

Pre-existing dengue antibodies could potentially exacerbate disease severity through antibody-dependent enhancement (ADE). Current serological assays focus on measuring neutralizing antibodies for vaccine evaluation, but don't measure sub-neutralizing antibodies that enhance infection via Fcγ receptors. Consensus on a standardized system for measuring dengue virus ADE remains elusive.

### Methods

In this study, we compared and evaluated ADE responses using two different methodologies in healthy blood donors (n = 12) and secondary dengue patients' (n = 12) samples with pre-existing IgG antibodies to dengue virus (DENV). We performed an ADE-infection assay in FcγRIIa-expressing U937, K562, and Vero-CD32a cells. Foci-reduction neutralization test (FRNT) was performed simultaneously in Vero and Vero-CD32a cells, and reduction in neutralization titres was examined in Vero-CD32a cells.

### Results

Out of 12 blood donors, all 9 anti-dengue IgG-positive donors demonstrated ADE through infection-enhancement assay against DENV-2 and DENV-4 serotypes in U937 and K562 cells, but not in Vero-CD32a cells. None of the anti-dengue IgG-negative donor samples exhibited ADE against DENV in all three cell lines. Fold-enhancement of DENV-2 infection was comparable in the two cell lines whereas, fold-enhancement of DENV-4 infection was significantly higher in K562 than in U937 cells. Comparable neutralizing antibody titres in Vero and Vero-CD32a

**Data availability statement:** All relevant data are within the manuscript.

**Funding:** This work was supported by DBT-BIRAC, India (Grant No. BIRAC/BT/NBM0095/02/18). The funders had no role in study design, data collection and analysis, the decision to publish, or the preparation of the manuscript.

**Competing interests:** The authors have declared that no competing interests exist.

cells against DENV-2 and DENV-4 serotypes suggest that donor samples did not exhibit any enhancing activity in Vero-CD32a cells. Comparable DENV-2 titres and significantly lower DENV-4 titres were obtained in Vero-CD32a than in Vero cells in secondary dengue patient samples, indicating that enhancing activity was influenced by DENV serotypes.

## Conclusion

In summary, infection-enhancement assay using K562 cells was superior to U937 and Vero-CD32a cells in evaluating ADE. Samples with high neutralizing activity demonstrated very low levels of infection-enhancing activity in Vero-CD32a cells. Comparison of FRNT titres in Vero and Vero-CD32a cells is not suitable for detecting ADE. Our findings suggest that infection-enhancing activities are apparent at sub-neutralizing concentrations of dengue virus antibodies in all individuals exposed to dengue virus.

---

## Introduction

Dengue is one of the most prevalent mosquito-borne viral diseases posing a continued threat to global health due to its impact on morbidity and mortality. Dengue affects at least 128 different countries, with about 390 million cases worldwide each year and an estimated 20,000 deaths annually [1]. A 2016 study estimates 50–100 million infections per year, with the highest dengue mortality among children in Southeast Asia [2]. Before 2009, dengue infection incidence rates in India were 6.34 per million people, rising sharply to 38.41 per million from 2010 to 2014 [3].

Dengue virus (DENV) infection is often self-resolving and is caused by four distinct dengue serotypes. Antibodies generated during primary DENV infection usually protect against the infecting serotype, however, this protection is limited against the other serotypes. When these antibodies wane out to sub-neutralizing levels, exposure to heterotypic virus infections leads to disease severity. This phenomenon termed as "antibody-dependent enhancement (ADE)" plays a significant role in the pathogenesis of severe dengue [4]. Cross-reactive or sub-neutralizing antibodies form an immune complex with heterotypic DENV to enter monocytes, macrophages, and dendritic cells through Fcγ receptors [5]. FcγRIIa or CD32a primarily facilitates the entry of the dengue virus through DENV-immune complexes, further triggering an enhanced viral egress process [6]. DENV-antibody complexes in secondary dengue patients are highly infectious and capable of producing higher levels of viremia when assayed with FcγR-expressing cells [7]. Due to this trait, FcγR-expressing cells proved effective in isolating dengue viruses from patients with secondary dengue infections [8]. These reports suggest the role of FcγRIIa-mediated signal transduction particularly the cytoplasmic domain is necessary for ADE to occur [9,10].

A recent modelling framework implies that homologous infections confer immunity while secondary heterologous infections augment virus replication [11]. A few original studies have demonstrated ADE assays by infecting either peripheral blood

leukocytes or human skin dendritic cells, and human macrophages with DENV in the presence of anti-DENV immune sera and measured infection enhancement in terms of plaque assay-based virus titration [12,13] or percent infected cells by flow-cytometry [14]. Different cell lines expressing FcγRs like U937 [15], K562 [16], and THP-1 [17] were recently used to measure the percent infected cells by flow cytometry. In later studies, a comparison of neutralizing antibody titres was assessed simultaneously in non-FcγR and FcγR-expressing cells [9,18–20]. Fold-enhancement was calculated to determine the enhancing activity in serum samples using only FcγR-expressing cells [20]. A heterotypic secondary dengue inoculation among marmosets produced high levels of viremia when the neutralizing antibodies were assayed with FcγR-expressing cells [21]. Alternatively, ADE infection assays were performed using FcγR-expressing cells and either focus-forming assay [22], or real-time polymerase chain reaction [23] were used to measure enhanced viral infection from the culture supernatants. Enhanced infection levels were also measured through circulating NS1 protein as a marker for viremia [24]. A simple automated colorimetric estimation of ADE using ELISA rather than traditional FRNT assays was shown to be effective in large-scale surveillance studies [25]. Recent advancements in measuring enhanced DENV infection involve the use of dengue reporter virus particles as a high-throughput method [26].

Developing a dengue vaccine is challenging due to the complex nature of the four antigenically distinct virus serotypes and the associated risk of ADE. Ensuring that vaccines induce protective immunity against all four serotypes without causing ADE is a key consideration in vaccine development and its evaluation in the population. Current serological methods continue to underscore ADE, emphasizing the critical need for reliable, reproducible, and affordable testing [27].

While studies determining dengue virus ADE stretch back to the 1960s, it is unclear how these findings upgrade clinical investigations or vaccine trials in actual practice. To date, different methodologies on ADE are overwhelming, with no conclusion on the best-suited methods for ADE measurement. In this study, we aimed to compare and evaluate two common methodologies used to measure the enhancement of infection. These include – a) infection enhancement-based ADE assay in FcγRIIa expressing K562, U937 and Vero-CD32a cells, b) ADE by foci-reduction neutralization test in non-FcγR expressing Vero and FcγRIIa expressing Vero-CD32a cells. Our findings suggest that infection-enhancement determined by K562-mediated ADE assay is a superior method than the comparison of neutralizing antibody titres in Vero and Vero-CD32a cells.

## Materials and methods

### Ethics approval

This study was approved by the Institutional Ethics Committee of Bharati Vidyapeeth Deemed University, Bharati Hospital & Research Centre, Pune (IEC/2019/33). The Ethics Committee permitted the use of leftover samples to standardize assays. The leftover samples collected in 2018 and stored at −80°C were used for the present study from 01/11/2023–30/06/2024. The laboratory used coded samples and had no access to information about blood donors and dengue patients.

### Samples

A total of 24 samples collected from 12 healthy blood donors (plasma samples) and 12 secondary dengue patients (serum samples) were included in this study. Out of 12 secondary dengue patients, 6 presented without warning signs, and 6 presented with warning signs. History of clinical dengue was not available for the blood donors.

### Serological diagnosis

All 24 samples were subjected to DENV NS1 enzyme-linked immunosorbent assay (ELISA, J Mitra, India, Cat no. IR031096), anti-DENV-IgM Capture ELISA (Panbio, Cat no. 01PE20), and anti-DENV-IgG indirect ELISA (Panbio, Cat no. 01PE30). A positive result in the DENV-Indirect IgG ELISA indicates that the sample has had prior exposure to dengue

viruses. To detect recent secondary dengue infection with elevated levels of IgG, anti-DENV-IgG Capture ELISA (Panbio, Cat no. 01PE10) was performed. This assay differentiates between primary and secondary dengue infections based on Panbio units. A cut-off value of > 22 units is used to identify secondary infections, which is calculated as per the manufacturer's instructions.

## Cells

U937 (human monocytic cell line, American Type Culture Collection, ATCC, USA) and K562 (human erythroleukemic cell line, NCCS, Pune) were cultured in 10% FBS containing RPMI 1640 medium (Sigma-Aldrich, St. Louis, MO, USA). Vero (CCL-81, African green monkey kidney epithelial cells, ATCC, USA) cells were cultured in minimal essential media (MEM) (Invitrogen, Carlsbad, CA), supplemented with 10% v/v heat-inactivated fetal bovine serum (FBS, Life Technologies, CA) and 1% penicillin-streptomycin (P/S) (Invitrogen). Vero-CD32a cell line, a kind gift by Dr. Stephen Whitehead, NIH, USA, was propagated in Opti pro-SFM media containing 10% v/v heat-inactivated FBS, 4mM L-glutamine (Invitrogen) and 0.2mg/mL geneticin (Invitrogen). A sequence encoding the human CD32a receptor was cloned into the mammalian expression vector pT-Rex DEST30 (Invitrogen). To generate stable cell lines, Vero cells were transfected with the CD32a plasmid, and a stable cell line was established by selection marker geneticin (G418) [28].

## Viruses

Virus stocks of Dengue serotype-2 (DENV-2, accession no. MW191699) and serotype-4 (DENV-4, accession no. MG272272) were propagated in Vero cells, the cell culture supernatant was harvested, aliquoted, and stored at −80°C for further use. For the time kinetics study, Vero and Vero-CD32a cells were infected with 0.1 multiplicity of infection (MOI) of virus stocks, and 200µl of the inoculum was added to 24-well plates. The plates were incubated at 37°C and with 5% $CO_2$ for virus adsorption for 2 hours. The cells were washed and supplemented with fresh 2% MEM and 2% Opti pro-SFM for Vero and Vero-CD32a cells, respectively. Supernatants were collected at 24-, 48-, 72-, and 96-hours post-infection, and a focus-forming assay was performed on a 96-well plate.

## Antibodies

The hybridoma, flavivirus cross-reactive D1-4G2-4–15 or HB112 (referred to as HB112 hereafter) was obtained from ATCC. HB112 hybridoma cells were grown in hybridoma SFM media (Cat no. 12309019, Gibco) containing 20% FBS and 1% P/S. HB112 hybridoma cells were depleted to serum-free conditions, and supernatants were collected by centrifugation at 1500rpm for 5 minutes. The resultant supernatants were concentrated using Amicon Ultra-15 tubes (Cat no. UFC903024, Merck Millipore, Milford, MA, USA) by centrifugation at 5000 x g. Then, the concentrated antibody was further purified using A/G columns from the NAb Spin kit (Cat no. 89950, Thermofisher Scientific, USA).

## Focus-forming assay (FFA)

Vero cells were seeded at 10,000 cells/ 100µl per well in a 96-well plate, one day before infection. The supernatants were ten-fold serially diluted, and 50µl of the inoculum in duplicate wells was added to the seeded plates. The plates were incubated at 37°C, 5% $CO_2$ for virus adsorption, and then overlaid with 100µl of 1% carboxymethylcellulose (Aquacide-II, Cat no. 17851, EMD Millipore, USA) containing 2% MEM. 48 hrs post-infection, the overlay media was aspirated, and the cells were washed with 1X PBS. The cells were fixed with 100µl of 3.7% formaldehyde solution and incubated at room temperature (RT) for 30 minutes. On rinsing with PBST, the cells were permeabilized with 0.2% Triton X-100 for 15 minutes. The plates were washed with PBST and blocked with 2.5% non-fat dried milk solution for 1 hour. On rinsing, pan-flavivirus 4G2 monoclonal antibody (MAb, HB112) as primary antibody was added to the cells and incubated for 2 hours. HRP-conjugated goat anti-mouse IgG secondary antibody was added to each well and incubated for 1 hour. Viral

foci were visualized after incubation with True-Blue Peroxidase substrate in the dark at RT for 30 minutes. The number of virus-infected foci was counted using the CTL ImmunoSpot machine (S6 Macro, CTL, USA), and virus titre was expressed as focus-forming units (ffu) per ml.

### Foci reduction neutralization test (FRNT)

Heat-inactivated serum/plasma samples were two-fold serially diluted, starting from 1:5–1:5120, and 120µl of this mixture was incubated with 120µl of DENV-2 and DENV-4 working stock, yielding 25–60 foci in Vero and Vero-CD32a cells. The resultant mixture was incubated at 37°C, 5% $CO_2$ for 1 hour, and 50µl was added to pre-seeded 96-well plates in duplicate wells and further incubated for 90 minutes. Overlay media was then added to Vero and Vero-CD32a cells. 48 hours post-infection, dengue virus-infected cells were immunostained as described earlier. $FRNT_{50}$ titres were expressed as the last serum dilution showing a 50% or greater reduction in foci counts as compared to virus control wells.

### ADE assay

K562 and U937 cells were seeded at densities of either 20,000 or 50,000 per well, on the day of infection, in 50µl of 2% RPMI media. Vero-CD32a cells were seeded at a density of 10,000 cells per well one day before infection. Pan-flavivirus 4G2 antibody (HB112) was tenfold serially diluted, ranging from 50 µg/ml to 0.005 µg/ml, and allowed to incubate with DENV-2 and DENV-4 viruses at MOI 5, 1, 0.2 and 0.04 for 1 hour. Heat-inactivated serum or plasma samples were ten-fold serially diluted from 1:10–1:100,000, and incubated with an equal volume of DENV viruses for 1 hour. The resultant mixture was added to the cells for adsorption. After 2 hours, cells were washed and resuspended in 100µl of infection medium containing 2% RPMI for K562 and U937 cells or 2% MEM for Vero-CD32a cells. Supernatants were harvested at 24hr, 48hr, and 72hr to observe the enhancement effect. 50µl of neat and ten-fold diluted supernatant was used to perform a focus-forming assay on Vero cells. Virus-infected cells were counted using CTL Immunospot machine. Fold enhancement values were determined using the following ratio: (mean foci count at different sample dilutions)/ (mean foci count in the absence of sample, no antibody control). A baseline was drawn at the sum of the mean of the negative control plus three times the standard deviation (SD) value obtained from three negative control samples and used as a cut-off to differentiate between enhancing and non-enhancing activity. Enhancing activity was defined as positive when values were greater than the cut-off [20]. A cut-off fold-enhancement value of 10ffu/ml and above was used to differentiate between infection-enhancing and non-enhancing activity.

### Simultaneous measurement of IgG-anti-DENV antibody positivity, neutralizing antibody, and fold enhancement of infection at sub-neutralizing antibody dilution

Plasma samples from 10 healthy donors were 10-fold serially diluted starting from 1:10–1:100,000 using 2% RPMI media, i.e., 60µl of sample was diluted in 540µl of RPMI. From the same set of serially diluted samples, an Indirect IgG ELISA (Panbio), a FRNT, and an ADE assay were setup in their formats. Indirect IgG and FRNT data were represented as positive/negative at each dilution. Enhancement of infection, i.e., fold enhancement was expressed as focus-forming units per ml (ffu/ml) obtained at a particular sub-neutralizing dilution.

### Data analysis

All statistical analyses and graphical representations were performed using GraphPad Prism Software version 10 (Graph-Pad Software Inc, San Diego, USA). The tests used for statistical analysis were indicated in the respective figure legends. The Wilcoxon signed-rank test was used to compare the data among the U937, K562 and Vero-CD32a cell lines. A paired t-test was used to compare FRNT50 titres obtained when using low and high-input virus control groups in Vero and Vero-CD32a cell lines. A non-parametric Mann-Whitney test was used to compare the geometric mean neutralizing antibody

titres among Vero and Vero-CD32a cell lines. Statistical significance in FRNT50 titres between Vero and Vero-CD32a cells in healthy donors and secondary dengue patients' samples was estimated by the Wilcoxon signed rank test. A p-value of ≤ 0.05 was considered significant.

## Results

To compare two test methods for the assessment of ADE response against DENV-2 and DENV-4 viruses, we used a panel of twenty-four serum/plasma samples representing healthy blood donors (n = 12) and acute secondary dengue patients (n = 12). Table 1 depicts the details of the samples used in the study. Only 3 healthy donors were negative for NS1 antigen and IgM/IgG antibodies to DENV, reflecting the high endemicity of the dengue virus.

### Fc-gamma receptor IIa (FcγRIIa) or CD32a expression in different cell lines

The expression of FcγRIIa in K562, U937, and Vero-CD32a cells was measured as the percentage of the anti-CD32a FITC-conjugated antibody-positive cell population by flow cytometry. Differential levels of CD32a expression were observed in K562, U937, and Vero-CD32a cells, while no expression was observed in Vero CCL-81 cells (Fig 1A). Established cell lines like K562 and U937 showed a higher expression of FcγRIIa i.e., 68.55% and 55.81%, respectively. The stable cell line Vero-CD32a exhibited a maximum expression of 43% at Passage 5 (P-5), after which the levels dropped to 34% by P-12 and 22% by P-15. From passages 15–25, the FcγRIIa expression level remained the same. Therefore, Vero-CD32a cells were used till P-12 (Fig 1B).

### Optimization of ADE infection assay for the measurement of enhanced viral titers

To standardize the ADE-infection assay, the pan-flavivirus cross-reactive MAb 4G2 (HB112) antibody known to enhance dengue virus infection of all four serotypes was used. Different parameters such as cell lines, cell density, hours post-infection, and virus multiplicity of infection were assessed.

**Effect of different cell lines.** We first evaluated antibody-dependent infection enhancement in different FcγRIIa-expressing cells, i.e., U937, K562, and Vero-CD32a cells. Vero cells were used as the non-FcγRIIa expressing cells (negative control). The peak antibody-dependent enhancement of the DENV-2 virus infection was estimated using different concentrations of HB112 antibody in all four cell lines. The peak enhancing concentration of HB112 was 1 µg/mL in both U937 and K562 cells. However, in Vero-CD32a and Vero cells, 'No Ab' control wells were fully saturated with infection at 1 MOI, and the distinction between infection and infection enhancement was unclear (Fig 2A). We further reduced the MOI to understand ADE response in Vero-CD32a cells as compared to Vero cells. At MOI 0.05, the 'No Ab'

**Table 1. Demographical and serological details of healthy blood donors and dengue patients used in the study.**

| Study Groups | Healthy donors (n = 12) | Acute Dengue (n = 12) |
|---|---|---|
| Median age (IQR)$ | 36 (29-41) | 24 (18-41) |
| Gender (Male/Female) | 12/0 | 6/6 |
| NS1+ | 0 | 1 (8.3%) |
| IgM+ | 0 | 7 (58.3%) |
| NS1 + IgM+ | 0 | 4 (33.3%) |
| IgG+ | 9 (75%) | 12 (100%) |
| Primary/ Secondary | NA* | Secondary |
| Post-onset date (POD) | NA* | 2-10 |

$IQR-Interquartile range;

*NA-Not applicable.

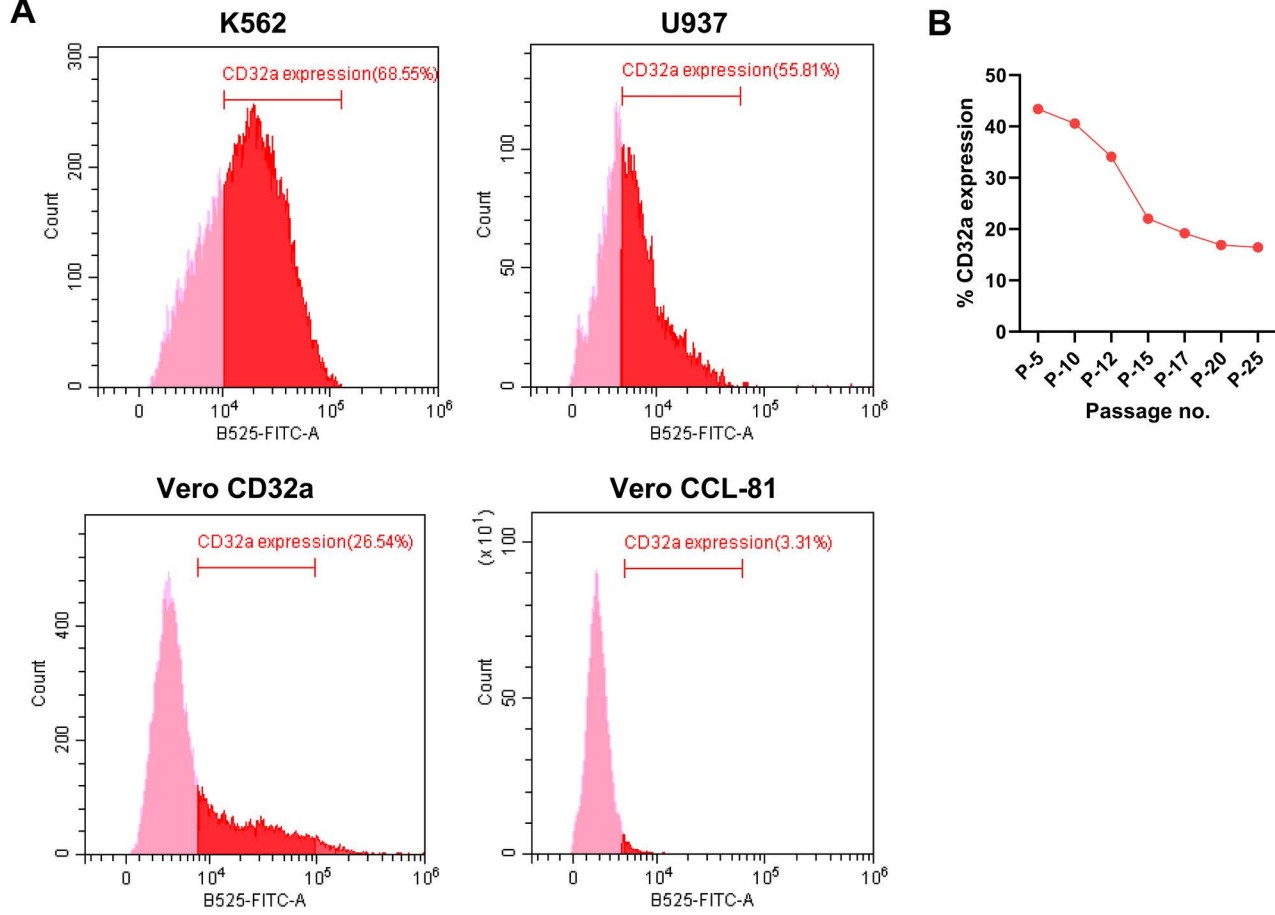

**Fig 1. (A) Representative flow cytometry plots showing the percentage of CD32a+ cells in K562, U937, Vero-CD32a, and Vero CCL-81 cells.** (B) A line graph of the percentage of CD32a+ cells indicates a decline in the expression of CD32a with an increase in the passage number of Vero-CD32a cells.

control wells exhibited countable foci in both Vero and Vero-CD32a cells; however, no distinct enhancement pattern was observed at any HB112 concentrations in Vero-CD32a cells compared to 'No Ab' control wells (Fig 2A). This may be due to sub-optimal levels of CD32a expression in Vero-CD32a cells. Thus, despite using lower MOI, ADE response could not be observed in the Vero-CD32a cells. Based on these observations, K562 and U937 cell lines were pursued for ADE infection assays.

**Effect of different cell densities.** The neutralization and enhancement of viral infection patterns were similar at both 20,000 cells/well and 50,000 cells/well seeding densities. At both cell seeding densities, 10 µg/mL of HB112 antibody concentration was insufficient to neutralize the DENV-2 virus (Fig 2B). In U937 cells, 50 µg/mL of HB112 showed complete neutralization of DENV-2, whereas K562 cells showed partial neutralization and were dependent on MOI used for DENV-2 infection (Fig 2C). U937 cells showed enhancement of DENV-2 infection at 0.5 µg/mL, whereas K562 cells showed between 5 to 0.5 µg/mL of HB112 antibody concentrations at the seeding density of 20,000 cells/well (Fig 2C).

**Effect of incubation time post-infection.** Here, DENV-2 virus at MOI 1 was incubated with 1 µg/mL of HB112 in U937 and K562 cells, and culture supernatants were collected at different hours post-infection. Depending on the incubation days post-infection, noticeable differences were observed in foci size and morphology (Fig 2D). At 24 hours post-infection,

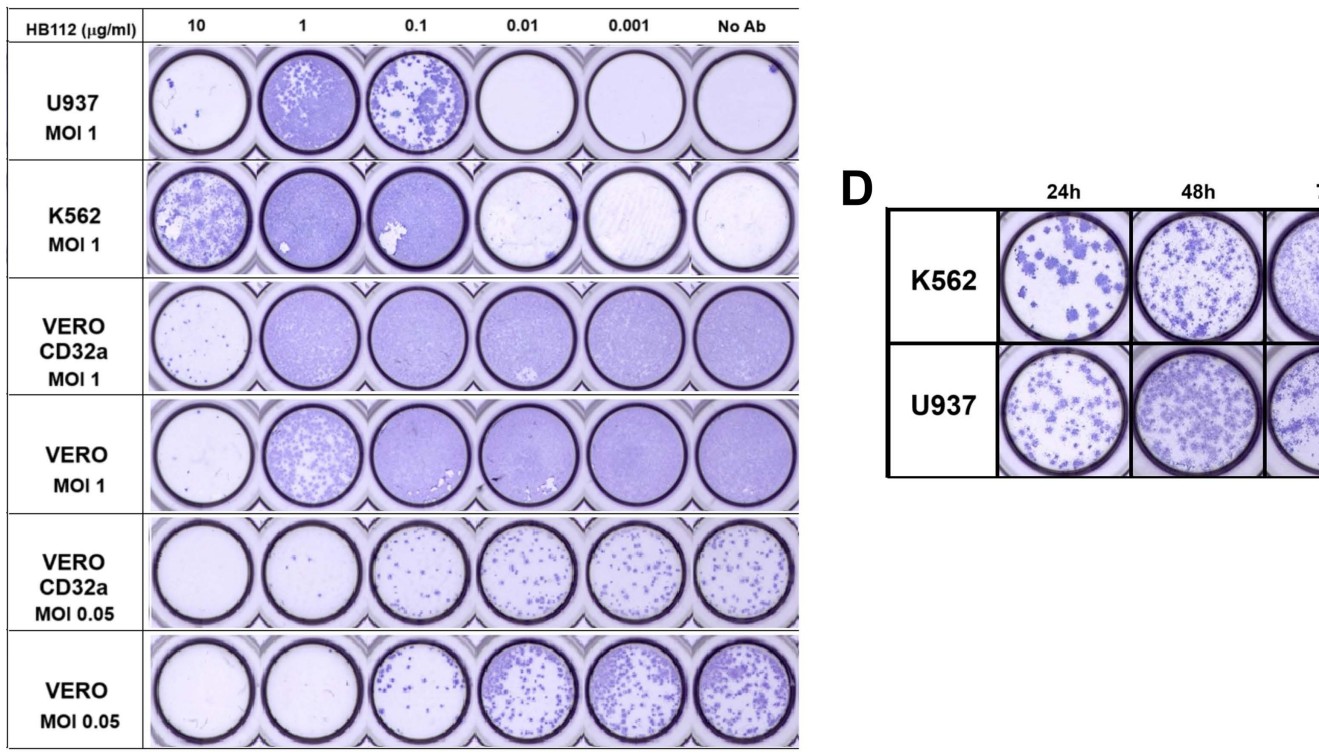

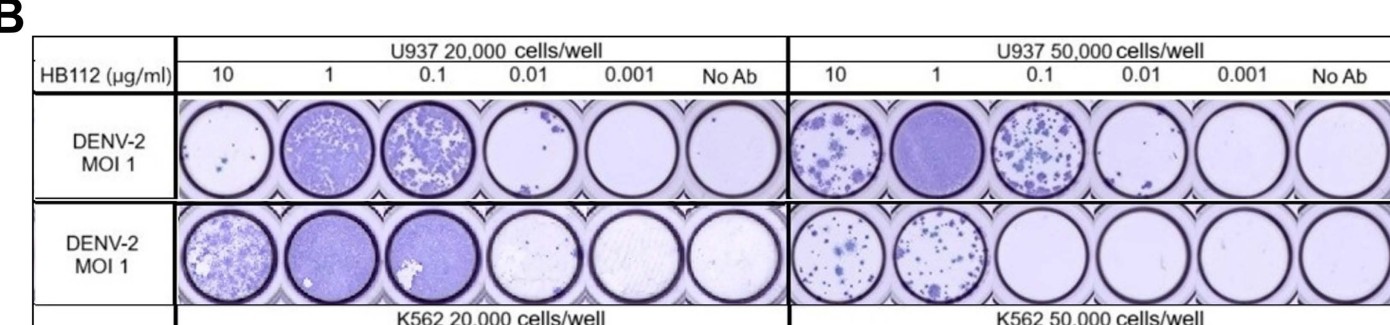

**Fig 2. (A) Comparison of infection enhancement patterns from supernatants collected from the wells incubating mixtures of different concentrations of HB112 antibody with DENV-2 virus.** HB112 antibody and DENV-2 virus at MOI 1 mixture were incubated with U937 and K562 cells for 24 hours. HB112 antibody and DENV-2 virus at MOI 1 and MOI 0.05 mixture were incubated with Vero-CD32a and Vero cells for 24 hours. (B) Comparison of infection enhancement patterns among different cell densities of 20,000 and 50,000 cells/well. Supernatants were collected from the wells incubating different concentrations of HB112 antibody with DENV-2 virus at MOI 1, incubated with U937 and K562 cells for 24 hours. (C) Comparison of infection

enhancement patterns in two cell lines, U937 and K562, seeded at 20,000 cells/well. Supernatants were collected from mixtures of different concentrations of HB112 antibody with DENV-2 virus at MOI 1 and MOI 0.2, incubated for 24 hours. (D) Comparison of the foci morphology at different time points. Supernatants were collected from mixtures of DENV-2 at MOI 1 and 1 µg/ml of HB112 incubated with K562 and U937 cells at 24-, 48-, and 72-hours post-infection.

the foci size was distinct and countable. However, as the hours post-infection progressed to 48 and 72 hours, infection rates increased significantly, and the foci increased in size with undefined borders and became less distinguishable.

**Effect of different input virus multiplicity of infection (MOI).** To check whether the input virus concentration has any effect on the enhancement of infection, we set up an experiment where a 10-fold diluted HB112 antibody, starting from 50 µg/ml to 0.005 µg/ml, was incubated with different MOIs - 5, 1, 0.2, and 0.04 of DENV-2 and DENV-4 viruses. As expected, a higher MOI of 5 yielded ~20 foci in no antibody (virus control) wells, and at a lower MOI of 0.04, no foci were seen in the no antibody wells in U937 cells. Relatively, fewer foci (less than 5) to zero foci were observed in the no antibody wells at higher and lower MOI in K562 cells (Fig 3). This further validates the lower susceptibility of U937 and K562 cells over Vero and Vero-CD32a cells. The enhancement of infection was only seen with virus-antibody complexes due to the presence of FcγRIIa on these cells.

The peak enhancement was seen at 0.5 and 5 µg/ml concentrations of HB112 antibody at different MOIs of DENV-2 virus in U937 and K562 cells, respectively. The degree of enhancement was much higher in K562 than in U937 cells at 1

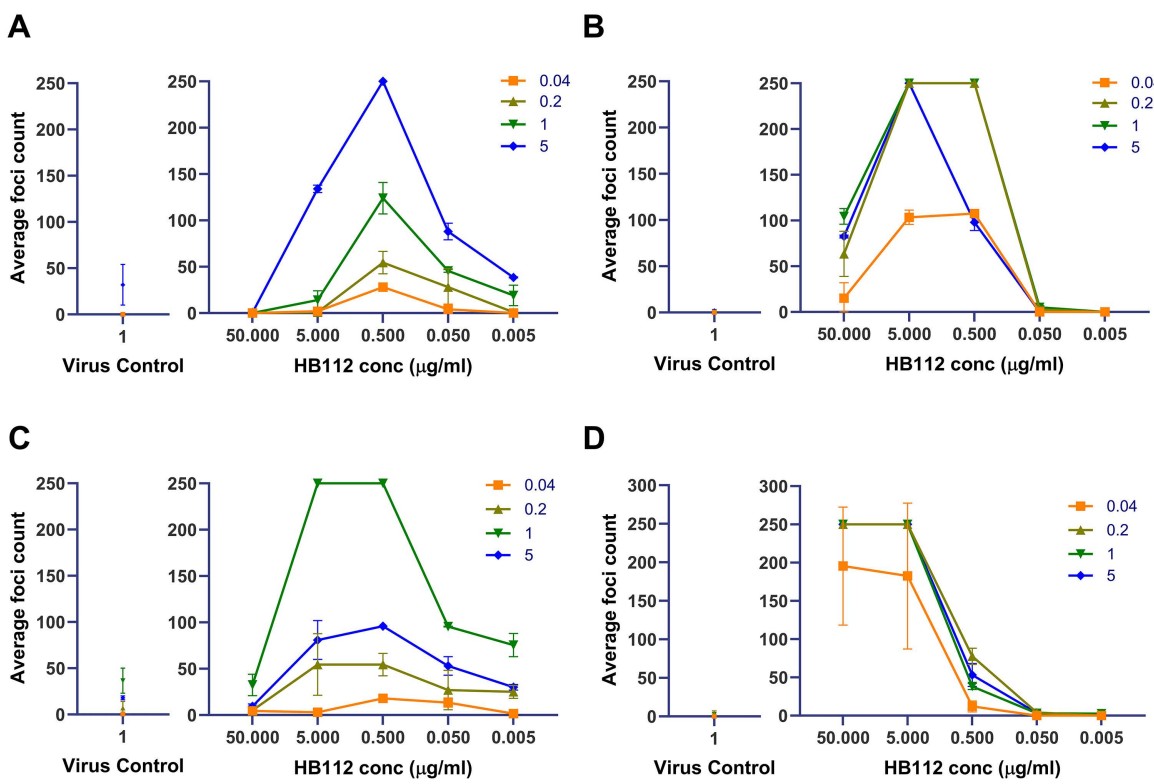

**Fig 3. A plot of average foci counts across concentrations of HB112 at different MOIs of DENV-2 and DENV-4 virus in both U937 and K562 cells.** A virus control panel indicates the number of foci obtained in virus control wells without any antibody mixture in the respective cells. The top panel indicates DENV-2 virus infection enhancement in (A) U937 and (B) K562 cells, whereas the bottom panel indicates DENV-4 virus infection enhancement in (C) U937 and (D) K562 cells.

and 0.2 MOIs of DENV-2 virus with 0.5 µg/ml of HB112 antibody (Fig 3A and 3B). For the DENV-4 virus, peak enhancement of infection was seen at the HB112 antibody concentrations of 5 µg/ml in both U937 and K562 cells. In K562 cells, even 50 µg/ml of HB112 antibody did not efficiently neutralize the DENV-4 virus at different MOIs. Like DENV-2, the degree of enhancement of DENV-4 infection was higher in K562 than in U937 cells, even at lower MOIs (Fig 3C and 3D).

**ADE infection assay using healthy donor samples**

The optimized ADE assay in K562, U937, and Vero-CD32a cells was used to screen twelve healthy blood donor samples at MOI 0.2 in K562 cells, MOI 1 in U937 cells, and MOI 0.05 in Vero-CD32a cells. All nine dengue IgG-positive donor samples showed peak fold enhancement of infection at different dilutions for DENV-2 and DENV-4 viruses, respectively, in K562 cells (Fig 4A and 4D) and U937 cells (Fig 4B and 4E). None of the donor samples exhibited any ADE activity in Vero-CD32a cells against DENV-2 (Fig 4C) and DENV-4 (Fig 4F) viruses. Rather, neutralization of donor samples was noted at dilution 1:20 when Vero-CD32a cells were used. All three dengue IgG-negative donor samples showed no enhancement of infection of DENV-2 and DENV-4 viruses in both K562 and U937 cells. The fold-infection enhancement was comparable when the DENV-2 virus was used in the two cell lines (p = 0.062, Fig 4G). For DENV-4, the fold-infection enhancement was significantly higher in K562 than in the U937 cells (p = 0.01, Fig 4H). Our data suggest a higher and more prominent enhancement of infection in K562 cells for both DENV-2 and DENV-4 viruses. Of importance, there were zero foci in the virus control of K562 cells as opposed to countable foci in the virus control of U937 cells, leading to clear results. Thus, K562 cells emerged as the most appropriate cell line for studying infection enhancement in the ADE assay.

**Optimization of foci reduction neutralization test (FRNT) parameters in Vero and Vero-CD32a cells**

We also attempted a comparison of the simultaneous measurement of $FRNT_{50}$ titres in Vero and Vero-CD32a cells. First, the replication kinetics of DENV-2 and DENV-4 were determined in both cells. The DENV-2 titre peaked at 48 hours in Vero cells and steadily declined to lower levels at 96 hours post-infection; however, in Vero-CD32a cells, the titre steadily increased and peaked at 72 hours post-infection, declining at 96 hours (Fig 5A). In the case of DENV-4, peak infectivity was observed at 72 hours and 96 hours post-infection in Vero and Vero-CD32a cells, respectively (Fig 5B). Our data clearly showed that the infectious virus titres in Vero-CD32a cells were significantly lower than in Vero cells. This could be due to the absence of the rigid virus-antibody complex requirement for an efficient infection in FcγRIIa-expressing Vero-CD32a cells.

We systematically optimized test parameters like cell density, incubation time post-infection, and input virus concentration for DENV-2 and DENV-4 in both cell lines. As shown in Fig 5C, distinct and countable foci were seen at 48 hours post-infection in Vero and Vero-CD32a cells for both serotypes.

To understand whether the input virus concentration plays a critical role in determining the neutralizing antibody titre of a given sample, we performed FRNT using low (20–30 foci) and high (45–70 foci) input of DENV-2 and −4 serotypes employing a panel of 21 anti-dengue IgG-positive samples. Comparable antibody titres were observed against DENV-2 and −4 in both Vero and Vero-CD32a cells, as shown in Fig 6. Based on these observations, it was concluded that an input virus concentration yielding 20–70 foci gave similar titres in both cells.

**Comparison of neutralizing antibody ($FRNT_{50}$) titres in Vero and Vero-CD32a cells among healthy blood donor samples**

DENV-2 and DENV-4 specific neutralization tests were performed simultaneously in both Vero and Vero-CD32a cells to determine the neutralizing and enhancing capacity of serotype-specific DENV-2 and DENV-4 antibodies. In principle, due to the enhancement of infection, the $FRNT_{50}$ titres are expected to be lower in FcγRIIa-expressing Vero-CD32a cells. The fold reduction of infection was calculated as the ratio of the mean $FRNT_{50}$ titre in Vero-CD32a to the mean $FRNT_{50}$ titre in Vero cells for a particular sample. All three anti-dengue IgG-negative donors did not show any response to DENV-2 and DENV-4 in either of the cell lines. Individual sample analysis documented that 3/9 (33%) dengue IgG-positive blood donor

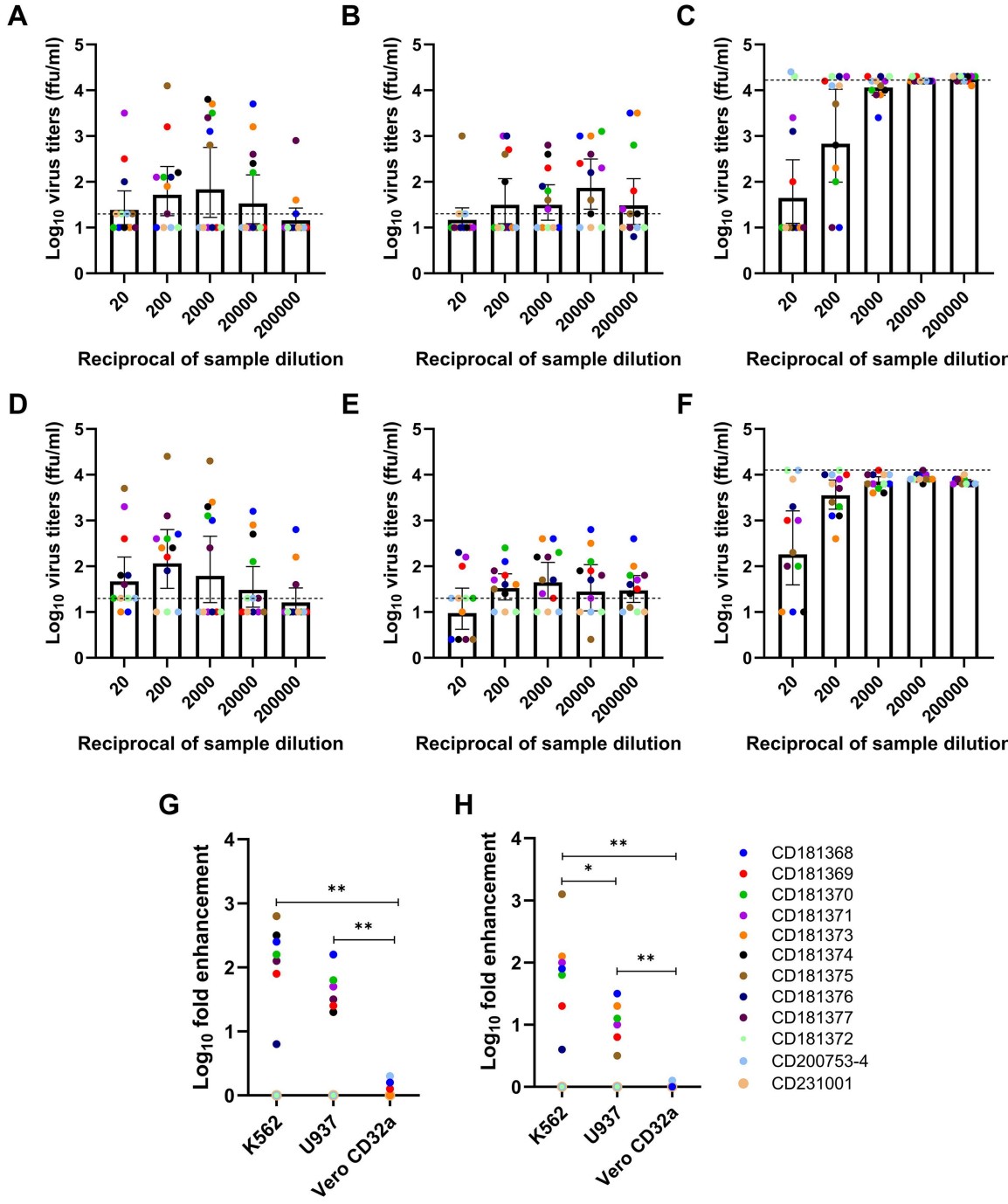

**Fig 4. A scatter plot of log₁₀ virus titers (ffu/ml) at different dilutions (1:20 to 1:20,000) of 12 healthy donor plasma samples.** Data is presented as geometric mean titers with 95% CI as error bars. The enhancement of infection for DENV-2 virus was observed in (A) K562, (B) U937, and (C) Vero-CD32a cells. The enhancement of infection for DENV-4 virus was observed in (D) K562, (E) U937, and (F) Vero-CD32a cells. The dotted line indicates the cut-off value above which fold enhancement was seen at different dilutions. At different sample dilutions, the virus titer was normalized by the average virus titer in virus control wells. A dot plot of the magnitude of fold enhancement in terms of log10 of virus titers, ffu/ml among 12 healthy donor samples for (G) DENV-2 and (H) DENV-4 viruses in K562, U937, and Vero-CD32a cells. The cut-off value was calculated from the mean values of three IgG-negative samples, and it was assigned a zero value on Y-axis since IgG-negative samples showed no fold-enhancement. The Wilcoxon signed-rank test was used for paired analyses. *indicates p-value <0.01, ** indicates p-value <0.001.

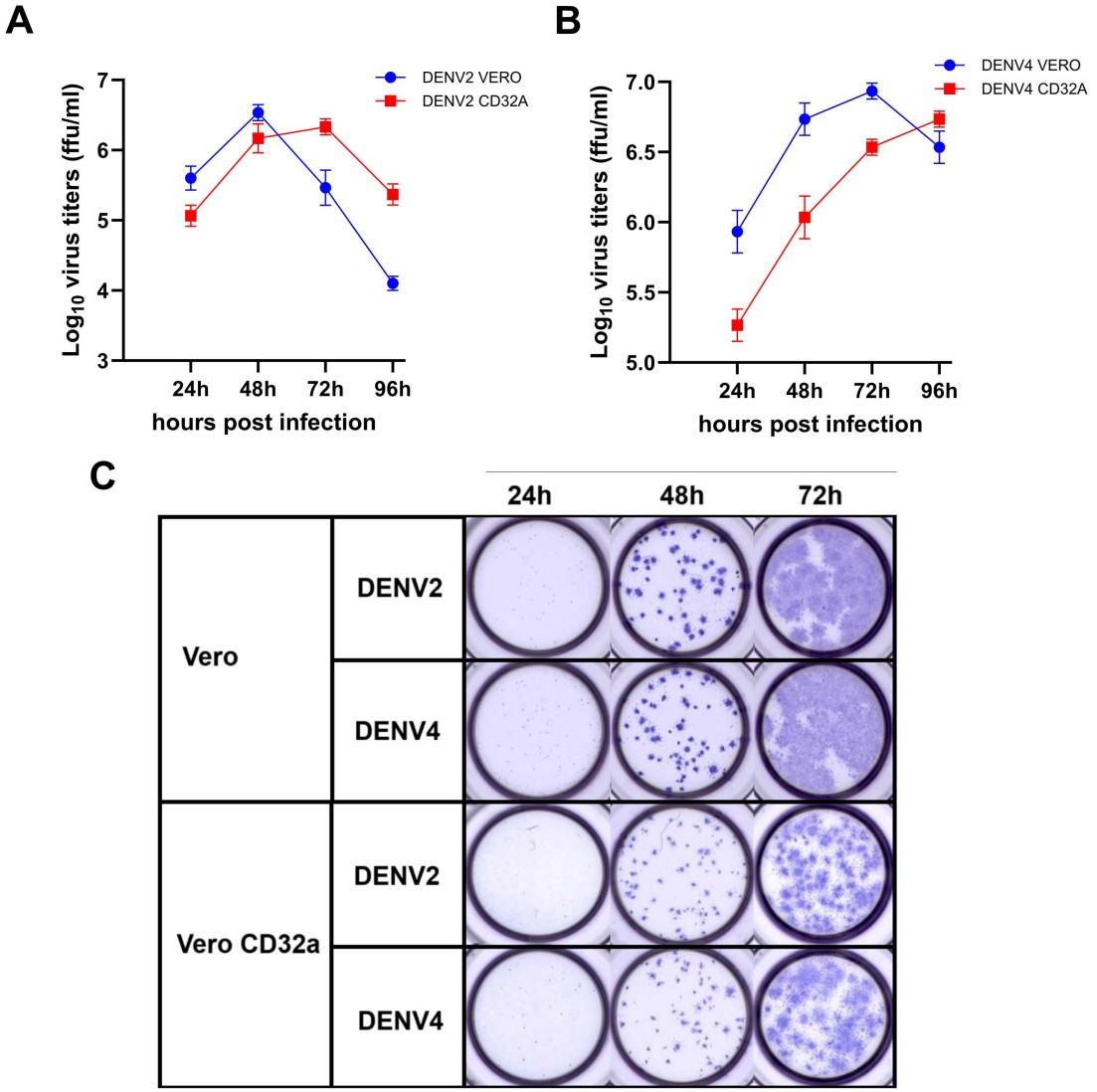

**Fig 5. Replication kinetics of DENV-2 & DENV-4 in Vero and Vero-CD32a cell lines.** Vero and Vero-CD32a cells were infected with (A) DENV-2 & (B) DENV-4 viruses at 0.1 MOI, and the culture supernatants harvested at different time points post-infection were analyzed for viral titers. Virus titers were determined by focus-forming assay on Vero cells. The line graph indicates the mean ± SD viral titers from three replicates. (C) The image compares the morphology of foci obtained in Vero and Vero-CD32a cells at different time points post-infection with DENV-2 and DENV-4 viruses.

samples showed a 0.4 to 0.5-fold reduction in $FRNT_{50}$ titres to DENV-2 virus in Vero-CD32a as compared to Vero cells (Fig 7A). However, the neutralizing antibody titres for DENV-2 virus was comparable in Vero (log10 GMT 3.06, 95% CI: 2.52–3.72) and Vero-CD32a cells (log10 GMT 2.93, 95% CI: 2.39–3.57; p = 0.23). In response to DENV-4, only one blood donor sample (11%) showed a 0.4-fold reduction in $FRNT_{50}$ titres in Vero-CD32a compared to Vero cells (Fig 7B). The $FRNT_{50}$ titres were comparable in Vero (log10 GMT 2.39, 95% CI: 1.86–3.09) and Vero-CD32a cells (log10 GMT 2.57, 95% CI: 2.09–3.14; p = 0.12) for DENV-4. The magnitude of enhancing activity in Vero-CD32a cells was minimal, up to 2-fold enhancement, and observed only in three and one donor samples, respectively, against DENV-2 and DENV-4 sero-types. Based on these results, it may be surmised that the donor samples with high neutralizing antibody titres possess negligible infection-enhancement activity in Vero-CD32a cells.

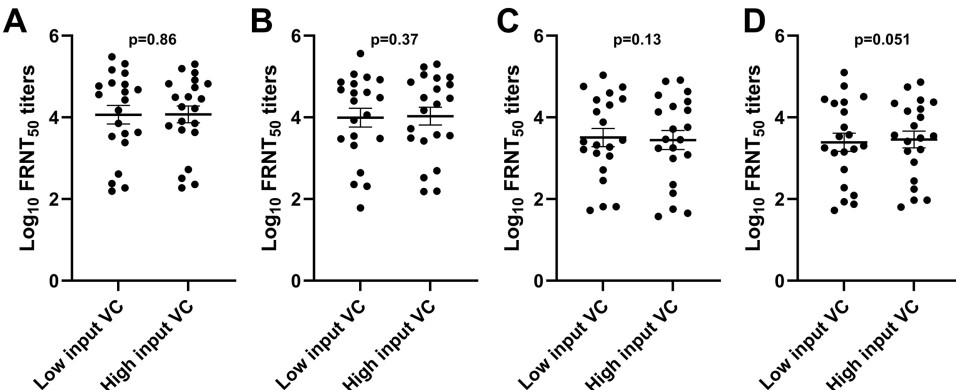

**Fig 6. Scatter plot of $log_{10}$ $FRNT_{50}$ titres of 21 samples (9 healthy blood donors and 12 secondary dengue patients) for (A & B) DENV-2 and (C & D) DENV-4 in Vero and Vero-CD32a cells, respectively.** Low input VC denotes an input of 500-700 ffu/ml, yielding 20-30 foci in virus control wells, and high input VC denotes an input of 1000-1200 ffu/ml, yielding 45-70 foci in virus control wells. A paired t-test was used for analyses.

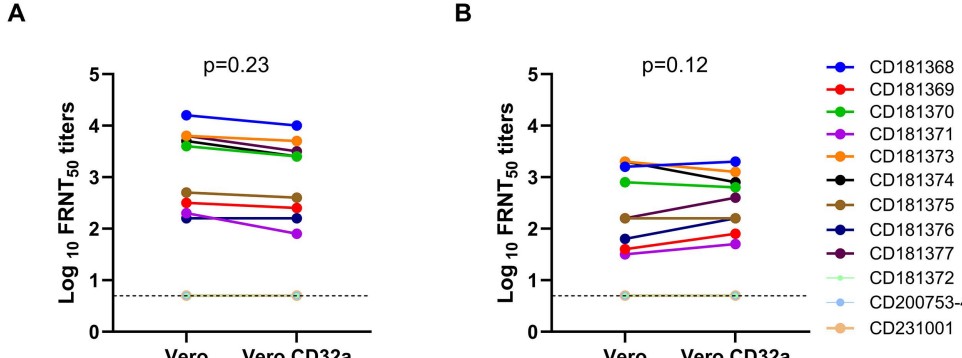

**Fig 7. A line graph of $log_{10}$ $FRNT_{50}$ titres of 12 healthy blood donor samples (9 dengue IgG-positive and 3 dengue IgG-negative samples) for (A) DENV-2 and (B) DENV-4 in Vero and Vero-CD32a cells.** The dotted line denotes a baseline showing geometric mean titres from three dengue IgG-negative samples. The data represents three independent experiments. Two-tailed P values were estimated by the Wilcoxon matched-pairs signed-rank test.

## Comparative evaluation of different test methods for the enhancement of infection among secondary dengue samples

As the mechanism of antibody-dependent enhancement is attributed to the severity of secondary dengue infection, we selected a panel of 12 secondary dengue patients' sera samples for comparison of the selected tests. These included (1) enhancement of virus infection using K562 cells and (2) simultaneous measurement of $FRNT_{50}$ titres in Vero and Vero-CD32a cells.

When the ADE assay was performed in K562 cells, all 12 secondary dengue samples showed infection-enhancing activity with peak enhancement at a particular dilution against DENV-2 and DENV-4 viruses. Most secondary dengue samples, i.e.,7/12 (58%) showed peak fold enhancement of infection at 1:200,000 dilution of the test sera against DENV-2 (Fig 8A) and DENV-4 viruses, in K562 cells (Fig 8B). The fold enhancement against both viruses was comparable in secondary dengue patient samples (Fig 8C). Notably, the peak enhancement was observed at higher dilutions in secondary

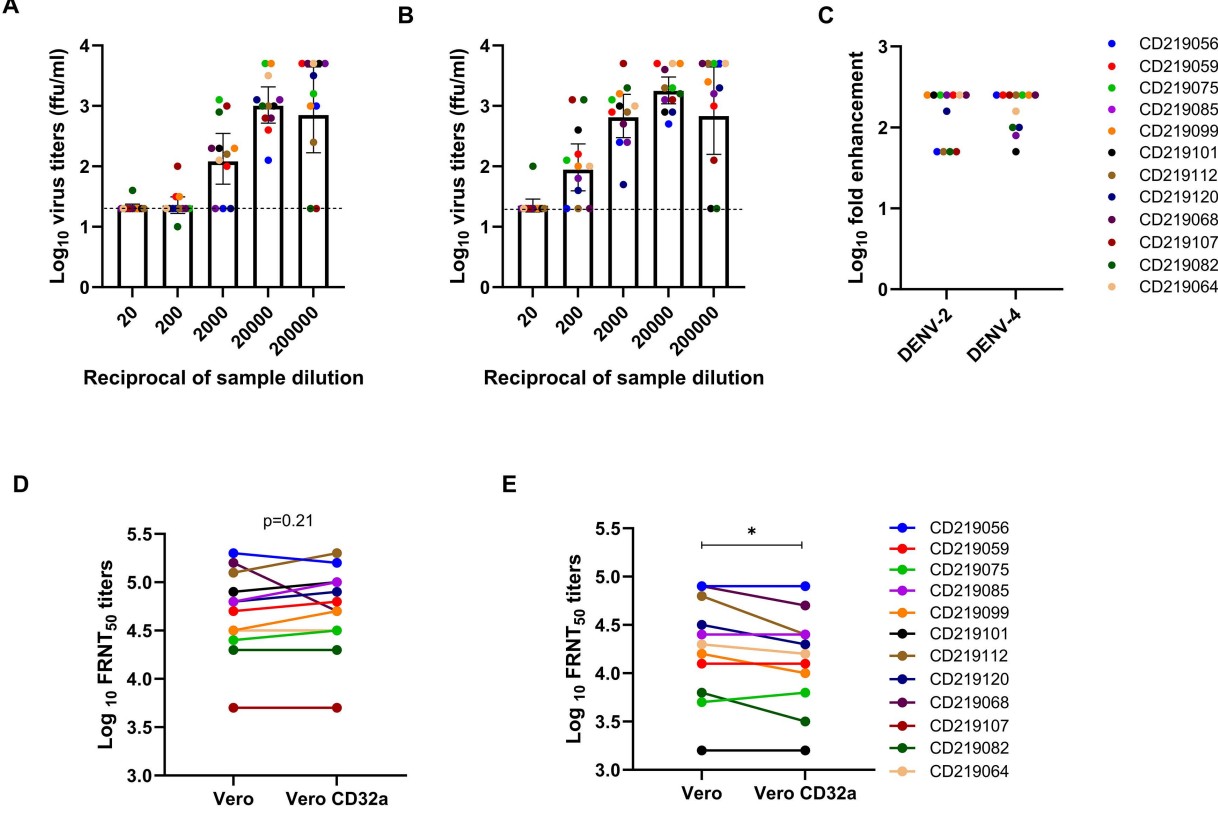

**Fig 8. A scatter plot of log₁₀ virus titers (ffu/ml) at different dilutions (1:20 to 1:20,000) of 12 secondary dengue patient samples.** The enhancement of infection for (A) DENV-2 and (B) DENV-4 viruses was observed in K562 cells. Data is presented as geometric mean titers with 95% CI as error bars. The dotted line indicates the cut-off value above which fold enhancement was seen at different dilutions. (C) Comparison of log 10 of fold enhancement of DENV-2 and DENV-4 infections in secondary dengue patient samples. A line graph of log₁₀ FRNT₅₀ titres of 12 secondary dengue samples for (D) DENV-2 and (E) DENV-4 in Vero and Vero-CD32a cells. Two-tailed P values were estimated by the Wilcoxon matched-pairs signed-rank test.

dengue patient samples as compared to healthy blood donor samples, probably due to high neutralizing antibody titres in secondary dengue patients.

As far as FRNT titres are concerned, being secondary infections, the FRNT₅₀ titres were high; for DENV-2 (log10 GMT 4.66, 95% CI: 4.38–4.96) in Vero and (log10 GMT 4.69, 95% CI: 4.42–4.99) in Vero-CD32a cells, and for DENV-4 (log10 GMT 4.13, 95% CI: 3.75–4.53) in Vero and (log10 GMT 4.02, 95% CI: 3.68–4.39) in Vero-CD32a cells. The FRNT₅₀ titres were comparable between Vero and Vero-CD32a cells for DENV-2 (p = 0.21), but significantly lower against DENV-4 in Vero-CD32a cells (p = 0.047). Individual sample analysis revealed that only one (8.3%, 0.4-fold reduction against DENV-2) and two (16.7%, 0.5-fold reduction against DENV-4) samples demonstrated very low enhancing activity in Vero-CD32a cells (Fig 8D and 8E). This data suggests that secondary dengue patient samples with high neutralizing antibody titres showed no to low infection-enhancing activity in Vero-CD32a cells, confirming the lack of utility of this test in detecting ADE.

100% ADE positivity in the infection-enhancement assay using K562 cells indicates that in all anti-dengue IgG-antibody-positive individuals, when antibodies decline to a particular sub-neutralizing concentration, ADE occurs upon exposure to a heterotypic virus. It resonates with the fact that protective antibodies in dengue-infected individuals decline to sub-neutralizing levels, where these antibodies, instead of protecting, cause antibody-mediated infection enhancement

on encounter with heterotypic dengue virus. This phenomenon will depend on the initial antibody titres, rate of decline, and timing of exposure to the heterotypic DENV to reach sub-neutralization levels of antibodies.

### Correlation between binding, neutralizing, and enhancing antibodies against all four dengue serotypes among healthy blood donor samples

Next, we evaluated the relationship between binding (ELISA), neutralizing (FRNT), and enhancing (K562-ADE) antibodies at sub-neutralizing antibody levels in a set of 10-fold serially diluted donor samples. We replicated the same ADE assay conditions for DENV-1 and DENV-3 viruses. For DENV-1 and DENV-3 FRNT assay, the same protocol was followed except for the change in incubation time after the addition of virus-antibody complex onto the Vero cells to 2 hours to achieve a more distinct foci pattern. As shown in Table 2, a few observations emerged: (1) All ELISA negatives were FRNT negative, but a majority did exhibit enhancing antibodies at varying levels, (2) Though ADE was seen at various dilutions, peak enhancement at a particular dilution was evident, which differed for different serotypes. Fold-enhancement of infection could be detected at a lower level even in the presence of serotype-specific neutralizing antibodies, (3) Peak enhancement differed for different serotypes and samples, depending on the titres of neutralizing antibodies. As expected, the dengue IgG antibody-negative sample, which had an equivocal result in ELISA, was negative for both neutralizing and enhancing antibodies against all four dengue serotypes at 1:20 dilution. In summary, in dilutions at which antibody enhancement was seen, samples had binding antibodies but were either depleted of (non-neutralizing) or had lower levels of (sub-neutralizing) neutralizing antibodies against all four serotypes.

## Discussion

In the present study, three FcγRIIa-expressing cell lines, K562, U937, and Vero-CD32a, were evaluated for the detection of ADE elicited in a given sample when tested at different dilutions. Our data suggests that an infection-ADE assay using the K562 cell line is the most appropriate method for the evaluation of ADE response.

We focused on the most widely reported FcγRIIa receptor-expressing cells for assessing ADE [29]. The levels of FcγRIIa in Vero-CD32a, U937, and K562 cells were demonstrated as a prerequisite to measure ADE. FcγRIIa receptor was most abundantly expressed in the K562, followed by U937 and the stable Vero-CD32a cell line (Fig 1). The results suggest that the inherent property of higher expression of FcγRIIa receptors on the surface of K562 cells determines the suitability of the cell line for the ADE assay.

Immune cells are the early targets of dengue virus infection. FcγR-bearing monocyte lineage cells are known to better reflect the *in-vivo* condition during dengue fever [30]. Initially, we used pan-flavivirus 4G2 (HB112) antibody as a positive control, which is known to enhance dengue virus infection [31]. Through optimization, we concluded that the antibody concentration and virus MOI were important determinants in ADE. Optimized test parameters were used to compare the enhancement of DENV-2 and DENV-4 infection in K562, U937, and Vero-CD32a cells. It was noted that higher virus MOI was required in U937 than in K562 cells for both DENV-2 and DENV-4 infection. Moreover, virus replication occurred in U937 cells even in the absence of antibodies, although at very low levels. However, virus replication was not observed in K562 cells even when infected with a higher MOI of both serotypes. In contrast, Vero-CD32a cells did not exhibit ADE activity at any of the HB112 antibody concentrations used. Studies have shown that U937 cells express both FcγRI and FcγRII, whereas K562 cells only express FcγRII [10,29]. FcγRIIa preferentially binds to IgG complexes and contains an immunoreceptor tyrosine-based activation motif (ITAM) in its cytoplasmic domain, whereas FcγRIIb transmits inhibitory signals through immunoreceptor tyrosine-based inhibitory motif (ITIM) in its cytoplasmic domain. It was demonstrated that in K562 cells, blocking FcγRIIa expression by either monoclonal antibody or siRNA treatment to knockdown FcγRIIa expression at mRNA level abrogated the ADE of DENV infection [10]. Co-expression of FcγRIIb in K562 cells significantly decreases the enhancement titre by 50%, suggesting that FcγRII isoforms regulate ADE activity in K562 cells. Moreover, the FcγRIIa receptor engineered to carry the ITIM motif abrogated ADE of DENV infection, suggesting that the cytoplasmic

**Table 2. Dilution-dependent correlation of binding (ELISA), neutralizing antibody positivity (FRNT), and fold-enhancement of serotype-specific DENV infection.**

| Sample ID | Sample Dilution | Indirect IgG | Neutralization in Vero* | | | | Fold enhancement of infection# | | | |
|---|---|---|---|---|---|---|---|---|---|---|
| | | | DENV-1 | DENV-2 | DENV-3 | DENV-4 | DENV-1 | DENV-2 | DENV-3 | DENV-4 |
| CD181368 | 1:20 | Positive | Y | Y | Y | Y | 200 | 0 | 160 | 40 |
| | 1:200 | Positive | Y | Y | Y | Y | 4000 | 20 | 18000 | 780 |
| | 1:2000 | Positive | N | Y | N | N | 7000 | 920 | 36000 | 2320 |
| | 1:20000 | Positive | N | N | N | N | 360 | 420 | 50000 | 15800 |
| | 1:200000 | Negative | N | N | N | N | 460 | 0 | 11200 | 1900 |
| CD181369 | 1:20 | Positive | N | Y | N | Y | 660 | 20 | 11600 | 340 |
| | 1:200 | Negative | N | N | N | N | 80 | 880 | 100 | 860 |
| | 1:2000 | Negative | N | N | N | N | 0 | 0 | 0 | 0 |
| | 1:20000 | Negative | N | N | N | N | 0 | 0 | 0 | 0 |
| | 1:200000 | Negative | N | N | N | N | 0 | 0 | 0 | 0 |
| CD181370 | 1:20 | Positive | Y | Y | Y | Y | 200 | 0 | 0 | 0 |
| | 1:200 | Positive | Y | Y | Y | Y | 5000 | 0 | 12000 | 1420 |
| | 1:2000 | Positive | N | N | N | N | 6400 | 740 | 16800 | 23200 |
| | 1:20000 | Negative | N | N | N | N | 240 | 0 | 260 | 500 |
| | 1:200000 | Negative | N | N | N | N | 0 | 0 | 0 | 0 |
| CD181371 | 1:20 | Positive | Y | Y | Y | N | 380 | 1380 | 1020 | 19600 |
| | 1:200 | Equivocal | N | N | Y | N | 460 | 1160 | 29400 | 17600 |
| | 1:2000 | Negative | N | N | N | N | 20 | 20 | 120 | 60 |
| | 1:20000 | Negative | N | N | N | N | 0 | 0 | 0 | 60 |
| | 1:200000 | Negative | N | N | N | N | 20 | 20 | 0 | 0 |
| CD181372 | 1:20 | Equivocal | N | N | N | N | 0 | 0 | 0 | 0 |
| | 1:200 | Negative | N | N | N | N | 0 | 0 | 0 | 0 |
| | 1:2000 | Negative | N | N | N | N | 0 | 0 | 0 | 0 |
| | 1:20000 | Negative | N | N | N | N | 0 | 0 | 0 | 0 |
| | 1:200000 | Negative | N | N | N | N | 0 | 0 | 0 | 0 |
| CD181373 | 1:20 | Positive | Y | Y | Y | Y | 40 | 0 | 0 | 40 |
| | 1:200 | Positive | Y | Y | Y | Y | 2800 | 20000 | 4000 | 980 |
| | 1:2000 | Positive | Y | Y | N | N | 6600 | 1000 | 40000 | 18000 |
| | 1:20000 | Equivocal | N | N | N | N | 2800 | 0 | 25000 | 16000 |
| | 1:200000 | Negative | N | N | N | N | 140 | 0 | 500 | 0 |
| CD181374 | 1:20 | Positive | Y | Y | Y | Y | 120 | 740 | 0 | 160 |
| | 1:200 | Positive | Y | Y | Y | Y | 5800 | 21600 | 9600 | 2980 |
| | 1:2000 | Positive | N | N | N | N | 7800 | 160 | 40000 | 18200 |
| | 1:20000 | Negative | N | N | N | N | 1800 | 0 | 10400 | 2600 |
| | 1:200000 | Negative | N | N | N | N | 100 | 0 | 80 | 300 |
| CD181375 | 1:20 | Positive | Y | Y | Y | Y | 1220 | 14800 | 5400 | 2460 |
| | 1:200 | Positive | N | N | N | N | 6400 | 20 | 40000 | 18800 |
| | 1:2000 | Negative | N | N | N | N | 3400 | 0 | 25000 | 2260 |
| | 1:20000 | Negative | N | N | N | N | 40 | 0 | 40 | 0 |
| | 1:200000 | Negative | N | N | N | N | 20 | 0 | 0 | 0 |

*(Continued)*

**Table 2.** (Continued)

| Sample ID | Sample Dilution | Indirect IgG | Neutralization in Vero* | | | | Fold enhancement of infection# | | | |
|---|---|---|---|---|---|---|---|---|---|---|
| | | | DENV-1 | DENV-2 | DENV-3 | DENV-4 | DENV-1 | DENV-2 | DENV-3 | DENV-4 |
| CD181376 | 1:20 | Positive | Y | Y | Y | Y | 60 | **820** | 1080 | **2240** |
| | 1:200 | Positive | Y | N | N | N | **4400** | 100 | **4400** | 1300 |
| | 1:2000 | Negative | N | N | N | N | 60 | 0 | 80 | 0 |
| | 1:20000 | Negative | N | N | N | N | 0 | 0 | 20 | 0 |
| | 1:200000 | Negative | N | N | N | N | 0 | 0 | 0 | 0 |
| CD181377 | 1:20 | Positive | Y | Y | Y | Y | 40 | 0 | 1100 | 640 |
| | 1:200 | Positive | Y | Y | N | N | **960** | 600 | **27600** | **22000** |
| | 1:2000 | Equivocal | N | Y | N | N | 340 | **1780** | 2100 | 2280 |
| | 1:20000 | Negative | N | N | N | N | 0 | 0 | 360 | 60 |
| | 1:200000 | Negative | N | N | N | N | 20 | 0 | 80 | 40 |

*The presence and absence of neutralizing activity are marked as Y-Yes and N-No.

#Highest fold enhancement as FFU/mL in ADE assay using K562 cells at a particular dilution is highlighted and marked in bold.

domain carrying ITAM motif of FcγRIIa determines ADE activity [10]. Higher DENV neutralization potency observed in U937 cells expressing both FcγRI and FcγRII receptors, further supporting the sub-optimal utility of U937 in studying ADE response [32]. Altogether, it implies that K562 cells might be more effective for measuring the ADE response (Fig 3).

Blood donor samples negative for anti-DENV IgG antibody showed no enhancement, whereas antibody-dependent enhancement of DENV-2 and DENV-4 infections in U937 and K562, but not in Vero-CD32a cells, was noted among all the antibody-positive donors. Thus, Vero-CD32a is not a suitable cell line for ADE assessment. The fold-enhancement of only DENV-4 infection was significantly higher in K562 than in U937 cells. However, the fold-enhancement of DENV-2 infection was comparable in the two cell lines, suggesting that enhancement patterns were influenced by DENV serotypes (Fig 4). Notably, the source of the virus was shown to have a profound impact on virus neutralization and ADE [22]. In the presence of viruses isolated from the same patients, sera from the acute secondary dengue patients had high levels of neutralizing activity and low ADE activity. However, no neutralization and high levels of ADE activity were observed in the presence of laboratory-adapted strains [22]. Evidence of ADE in all the maternal samples at different dilutions was reported using K562 cells in the Vietnamese cohort [33] and in Brazilian infants [34]. Even during the *in-vitro* evaluation of different vaccine formulations, all immunized mice sera showed ADE response at sub-neutralizing dilutions using K562 cells [35,36]. Taken together, studies have shown that 100% of the tested anti-DENV IgG-positive samples induce ADE at a sub-neutralizing dilution, and our results align with these reports.

The use of immune cells for ADE assay requires a surrogate plaque assay for estimating the increase in the number of viral particles. To overcome this obstacle, modified cell lines such as FcγR-expressing BHK-21 cells, CV-1, or COS-7 cells were established. The levels of viremia in dengue patients with secondary infection were detected ~10 times higher in FcγR-expressing BHK-21 cells than in FcγR-negative BHK-21 cells. However, when the samples from primary dengue patients were used, virus titres were comparable between FcγR-expressing and FcγR-negative BHK-21 cells. Thus, dengue virus-antibody complexes present in secondary dengue patients could be detected using FcγR-expressing cells mimicking *in-vivo* target cells [7]. A human anti-DENV serum from a DENV-3-infected patient showed enhanced DENV-1 and DENV-2 infection at a dilution of 1:1000 and 1:10000, respectively, when FcγRIIa-transfected COS-7 cells were used [9]. We also observed enhancement of infection at higher dilutions against DENV-2 and DENV-4 serotypes, although the infecting serotypes in secondary dengue patients were not known. Also, serum samples from infected monkeys demonstrated high levels of neutralizing antibodies to homotypic virus and enhancing activity at dilutions of 1:100–1:1000 against

homotypic virus and at dilutions of 1:10–1:100 against heterotypic virus [37], suggesting that serum samples demonstrate both neutralizing and ADE activities to a variable extent against homotypic and heterotypic infections.

Mammalian cell lines such as Vero and BHK-21 cells are the most commonly used cell lines for dengue virus neutralization assays [38]. Here, we optimized the test parameters of the FRNT technique in a 96-well plate using Vero and Vero-CD32a cells against DENV-2 and DENV-4 viruses. The history of clinical dengue was not known for blood donors; however, the high levels of neutralizing antibody titres confirm prior exposure to DENV. The overall neutralizing antibody titres were comparable in Vero and Vero-CD32a cells against DENV-2 and DENV-4 serotypes. It is known that samples with high neutralizing activity possess no or only low levels of infection-enhancing antibodies, as reported by us in this study as well as by others [19,20]. Our observations stand in stark contrast to other reported findings involving simultaneous measurement of neutralizing antibody titres using BHK-21 vs FcγRIIa expressing-BHK-21 cells and Vero vs CV-1-Fc cells [9,18,20,39]. The neutralizing antibody titres determined by FcγRIIa expressing-BHK-21 cells were lower than those obtained by using BHK-21 for all four serotypes. This could be due to (1) differential expression of FcγRIIa receptor on Vero-CD32a and FcγRIIa-expressing BHK-21 cells, and (2) the concentration of neutralizing antibodies in the samples. It is known that levels of neutralizing antibodies were inversely associated with infection-enhancement activities. Our donor samples exhibited high titres and were positive for neutralizing antibodies against ≥ 3 DENV serotypes. It has been shown that samples with multitypic neutralizing activity to ≥ 3 DENV serotypes have significantly lower enhancing activity [20]. In another study, samples from dengue patients with unknown dengue exposure history had significantly lower $PRNT_{50}$ titres in CV-1-Fc cells than Vero cells for each of the four serotypes except DENV-2 [18]. We noted comparable $FRNT_{50}$ titres in Vero and Vero-CD32a cells for DENV-2 in samples of secondary dengue patients (Fig 8D). Significantly lower DENV-4 $FRNT_{50}$ titres were observed in Vero-CD32a than in Vero cells (Fig 8E). Again, the cell lines used, CV-1-Fc versus Vero-CD32a, in the two studies were different. Moreover, enhancing activity may vary among serotypes. As shown by Moi et al (2012), samples with high neutralizing antibody titres against DENV-1 serotype showed lower enhancing activity against DENV-1 serotype and higher enhancing activity against DENV-4 serotype [20]. In contrast, samples from secondary dengue-infected Cambodian children showed enhancing activity against the infecting serotype DENV-2 but not against DENV-1, DENV-3, and DENV-4 despite having the highest neutralizing antibody levels against DENV-2 [15]. The neutralizing antibodies and enhancing antibodies vary between primary and secondary dengue patients but were not correlated with disease severity, suggesting that binding antibodies may contribute to pathogenesis via mechanisms other than neutralization or enhancement of infection [15].

In our study, blood donors and secondary dengue patients exhibited very high neutralizing antibody activity for DENV-2 and DENV-4 in Vero and Vero-CD32a cells. Therefore, immune enhancement at 1:10 diluted serum/plasma appears unlikely. In contrast, Moi et. al (2012) demonstrated the infection-enhancing activity at 1:10 serum dilution in 26% and 73% of samples to DENV-2 and DENV-4 serotypes, respectively, using FcγRIIa expressing-BHK21 cells [20]. In a single study, neutralizing and enhancing antibody responses were measured in longitudinal serum samples at day 208 and 10 years post-dengue vaccination using Vero and Vero-CD32a cells. At day 208 post-vaccination, all samples had high neutralizing antibody titres and no enhancement of infection. At 10 years post-vaccination, enhancement of infection was observed for DENV-2, −3, and −4 serotypes only in vaccinee sera who received yellow fever vaccines, and notably, at higher dilutions [28]. This study indicates that flavivirus-naïve participants who received the dengue vaccine did not demonstrate enhancement of infection in Vero-CD32a cells against any of the four DENV serotypes even after 10-years post-vaccination [28]. Samples with pre-existing flavivirus antibodies showed enhancement of infection, similar to dengue-infected patients' sera, showing enhanced Zika virus infection *in-vitro* [40]. Tick-borne encephalitis virus (TBEV) vaccine recipients develop a poor neutralizing antibody response against the Yellow fever vaccine due to the presence of pre-existing flavivirus immunity. In addition, the skewed IgG response towards the pan-flavivirus fusion loop epitope with potential to enhance antibody-mediated enhancement of dengue and zika virus infections among TBEV vaccinated individuals is noteworthy

[41]. Such studies underscore the importance of cautious interpretation of ADE response using modified FcγR-expressing cells. Multiple factors, such as neutralizing antibody titres, time of sample collection post-infection, and pre-existing flavivirus immunity, influence ADE response in modified FcγR-expressing cells. In addition, the requirement of matching the assay conditions and parallel testing of samples in a standard and a modified cell line complicates the test method. Our data emphasized on the superiority of using a single cell line, K562, in evaluating ADE response, averting all these factors.

Our data suggest that ADE occurs at sub-neutralizing antibody levels and fold-enhancement varied among DENV serotypes (Table 2). In a concurrent dengue case where the patient's serum sample was positive for both DENV-1 and DENV-4 serotypes, cross-reactive pooled serum samples neutralized the dominant DENV-1 serotype, enhancing the infection of subdominant DENV-4 serotype in FcγRIIa-expressing BHK cells [42]. Additionally, many ADE studies were restricted to a single serotype, which prevented a comprehensive understanding of ADE against all four serotypes [43]. It would be interesting to study the competing role of serotype-specific antibodies in ADE.

Our study has some limitations. Due to the unavailability of samples during the acute and convalescent phases, the infecting serotypes could not be determined in the secondary dengue patients. Identification of the infecting serotype(s) in DENV antibody-positive healthy blood donors was difficult, as the antibody responses were cross-reactive against the four serotypes. ADE is attributed to either cross-reactive antibodies or to antibodies targeting the fusion loop of envelope protein, prM surface protein, or to the maturation status of the virus. The presence of low to intermediate titres of pre-existing dengue antibodies correlated with increased risk of severe dengue in secondary-infected children [44]. Additionally, high titres of cross-reactive neutralizing antibodies are associated with protection from disease upon a second DENV infection [44]. Even host genetic factors such as genetic variants in FcγRIIa, DC-SIGN, MHC genes, MBL2, CCL-2, TNF-alpha, etc., contribute towards the progression to severe dengue. A single-point mutation A>G (rs1801274) in FcγRIIa, replacing histidine (H) at position 131 with arginine R, resulted in altered affinity towards IgG subclasses. HH homozygotes interact more efficiently with IgG2, while RR showed more affinity for IgG1 and IgG3 subclasses. The presence of H allele mediates inefficient binding of IgG1/IgG3 antibodies from dengue immune complexes, favouring ADE and severe infection. An association of FcγRIIa polymorphism (R131H) with the clinical outcome of disease in dengue patients was reported from Pakistan [45]. HH genotype presented increased DENV susceptibility towards severity. These findings were consistent with studies in Vietnamese and Cuban populations, but contrasting reports were noted in the Mexican population. Moreover, the R allele was associated with thrombocytopenia in Indian patients with dengue infection [46].

In summary, our study demonstrated that the infection-ADE assay using K562 cells is the most suitable method for detecting ADE during the evaluation of dengue vaccines in clinical trials or population-based studies. When samples from hyperendemic areas are screened, ADE seems universal among dengue IgG antibody positives, differing in dilutions of peak enhancement across serotypes of dengue virus. Our data strongly suggest that comparisons between neutralizing antibody titres using the FRNT method with non-FcγR expressing (Vero) and FcγR expressing (Vero-CD32a) cell lines can be misleading if samples with high and multitypic neutralizing antibody titres are used. The use of a single dilution for mass screening is not practical in such settings with high antibody titres. The infection-ADE assay in K562 cells can be automated for high-throughput.

## Acknowledgments

We wish to thank the staff of Translational Virology and National Immunogenicity and Biologics Evaluation Center Labs, Interactive Research School for Health Affairs (IRSHA), Bharati Vidyapeeth (Deemed to be University) for their support. We thank Dr. Stephen Whitehead, National Institute of Health (NIH), Bethesda, Maryland, USA for providing Vero-CD32a cell line. We sincerely thank Dr. Anna Durbin, John Hopkins Bloomberg School of Public Health, Baltimore, Maryland, USA for sharing the protocol of maintenance and sub-culturing of the Vero-CD32a cell line.

## Author contributions

**Conceptualization:** Akhilesh Chandra Mishra, Vidya Arankalle, Shubham Shrivastava.

**Data curation:** Shweta Chelluboina.

**Formal analysis:** Shweta Chelluboina, Shubham Shrivastava.

**Funding acquisition:** Akhilesh Chandra Mishra.

**Methodology:** Shweta Chelluboina, Darshan Kshirsagar, Gauri Panzade.

**Software:** Shweta Chelluboina.

**Supervision:** Shubham Shrivastava.

**Writing – original draft:** Shweta Chelluboina, Shubham Shrivastava.

**Writing – review & editing:** Akhilesh Chandra Mishra, Vidya Arankalle, Shubham Shrivastava.

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
