## [Decision Letter · Decision Letter 0]

29 Nov 2024

PONE-D-24-39295Comparative evaluation of methods for the measurement of antibody-dependent enhancement of dengue virus infection in FcγRIIa expressing cell linesPLOS ONE

Dear Dr. Shrivastava,

Thank you for submitting your manuscript to PLOS ONE. After careful consideration, we feel that it has merit but does not fully meet PLOS ONE’s publication criteria as it currently stands. Therefore, we invite you to submit a revised version of the manuscript that addresses the points raised during the review process.

1. Please carefully revise your manuscript according to the reviewers' comments and suggestions.2. Please double-check your manuscript, and address the following issues:(1) To find expert or company to polish the English of your manuscript.(2) To avoid missing text (for example, in line 43, "cell" missing after "Vero" and "Vero-CD32a").

(3) The results section of the abstract did not indicate that K562 cells were superior to U937 cells in evaluating ADE.(4)  In fact, a large number of studies have compared and evaluated the advantages and disadvantages of various cell lines in evaluating ADE. Please make appropriate citations in the article and compare them with your manuscript.(5) The title of the article does not match the content of the evaluation of ADE and neutralization experiments on different cell lines.(6) Plaque Assay (PA) should be replaced with Focus-forming assay (FFA). Virus titer should be expressed as focus-forming unit (FFU) per ml rather than PFU. Why is a culture medium containing aquacide used to cover cells in this experiment? Generally, a culture medium containing methylcellulose or low melting point agar is used to cover cells.(7) The pan-favivirus 4G2 antibody was used in the ADE assay, but the antibody HB112 appears in the experimental results section and figures. ==============================

We look forward to receiving your revised manuscript.

Kind regards,

Jinsheng Wen

Academic Editor

PLOS ONE

Additional Editor Comments (if provided):

Reviewers' comments:

Reviewer's Responses to Questions

**Comments to the Author**

1. Is the manuscript technically sound, and do the data support the conclusions?

Reviewer #1: Partly

Reviewer #2: Yes

2. Has the statistical analysis been performed appropriately and rigorously? 

Reviewer #1: Yes

Reviewer #2: Yes

3. Have the authors made all data underlying the findings in their manuscript fully available?

Reviewer #1: No

Reviewer #2: Yes

4. Is the manuscript presented in an intelligible fashion and written in standard English?

Reviewer #1: Yes

Reviewer #2: Yes

5. Review Comments to the Author

Reviewer #1: This manuscript written by Chelluboina et al. describes a basic study on the evaluating ADE activity against dengue virus. Authors compared some protocols of ADE assay using several Fcγ receptor-expressing cell lines (K562, U937 and Vero-CD32a cells). They also compared neutralizing activities between normal Vero and Vero-CD32a cells. They concluded that K562 cells were the most suitable to be used for evaluating ADE activity against dengue virus, and Vero-CD32a cells showed significantly lower neutralizing activity than normal Vero cells when DENV-2 was used as an assay antigen. Possibly the experimental design was elementary, thus the data were not exciting like I have ever seen before. I could not find any unique points which differentiated from other studies. Specific comments are below.

Major comments

Lines 132-135: Information of the Vero-CD32a cells is needed further. Please add a reference describing the origin of Vero-CD32a cells.

Lines 256-257 and Fig. 2A: Authors should repeat the ADE assay by reducing viral tier in the control.

The ADE pattern of dose response curves in Figs. 3A and 3B did not match the micrographs in Figs. 2A and 2B, although there were some differences in the concentration of HB112. Since the gap will make it difficult for readers to understand the ADE assay, these data should be correlated.

Figs. 4E and 4F suggested that K562 cells likely showed higher fold enhancement than U937 cells. However, micrographs in Figs. 2A and 2B indicated that U937 cells clearly displayed stronger ADE (fold enhancement) than K562 cells, which suggests that U937 cells may be more sensitive to ADE assay compared with K562 cells. This phenomenon is opposite to Figs. 4E and 4F. What is the cause of this difference? Did the difference between serum and MAb make this? Authors need to consider these points.

Minor comments

How many cells (cells/well) did you use in the assays of Fig. 2A?

“CELLS/ML” should be “cells/well” in Fig. 2B

Reviewer #2: 1. The manuscript titled "Comparative evaluation of methods for the measurement of antibody-dependent

enhancement of dengue virus infection in FcγRIIa expressing cell lines" demonstrate the utilization of different cell lines for ADE analysis. The title itself is little misleading. Actually, the authors have performed similar methodology using different cell lines, so it may be corrected.

2. In the manuscript, it is concluded that the infection-ADE assay using K562 cells is the most suitable

method to determine the enhancement of infection at sub-neutralizing dilutions, however it is directly corelated with the amount of FcγRIIa expression in the cells, which is quite obvious.

3. The cross-reactivity should be analyzed with all the four DENV serotypes or at least using VLPs.

4. Discussion part of the manuscript is written well, however, Abstract, Introduction, methodology, and result sections need revision in terms of clarity.

6. PLOS authors have the option to publish the peer review history of their article (what does this mean? ). If published, this will include your full peer review and any attached files.

**Do you want your identity to be public for this peer review?** For information about this choice, including consent withdrawal, please see our Privacy Policy .

Reviewer #1: No

Reviewer #2: No

---

## [Author Response · Author response to Decision Letter 1]

20 Jan 2025

Dear Editor and Reviewers,

Thank you for your valuable feedback on our manuscript titled "Comparative evaluation of methods for the measurement of antibody-dependent enhancement of dengue virus infection in FcγRIIa expressing cell lines" (Manuscript ID: PONE-D-24-39295 PLOS ONE). We appreciate the time and effort you have invested in reviewing our work. We have carefully considered all the comments and have made significant revisions to address them.

Response to the Academic editor:

1. Please carefully revise your manuscript according to the reviewers' comments and suggestions.

Response: We have addressed each comment and revised the manuscript accordingly.

2. Please double-check your manuscript, and address the following issues:

(1) To find expert or company to polish the English of your manuscript.

Response: We have tried to improvise our writing language for better clarity.

(2) To avoid missing text (for example, in line 43, "cell" missing after "Vero" and "Vero-CD32a").

Response: We have added missing text throughout the manuscript.

(3) The results section of the abstract did not indicate that K562 cells were superior to U937 cells in evaluating ADE.

Response: We have revised the results section of the abstract highlighting the superiority of K562 cells over U937 cells in evaluating ADE.

(4) In fact, a large number of studies have compared and evaluated the advantages and disadvantages of various cell lines in evaluating ADE. Please make appropriate citations in the article and compare them with your manuscript.

Response: We have compared our results with published literature by evaluating the different Fc�R expressing cell lines in the revised manuscript (Line no. 566 – 582, page no. 26).

(5) The title of the article does not match the content of the evaluation of ADE and neutralization experiments on different cell lines.

Response: We've made changes to the title.

(6) Plaque Assay (PA) should be replaced with Focus-forming assay (FFA). Virus titer should be expressed as focus-forming unit (FFU) per ml rather than PFU. Why is a culture medium containing aquacide used to cover cells in this experiment? Generally, a culture medium containing methylcellulose or low melting point agar is used to cover cells.

Response: Several papers in the literature make use of both plaque assay (PA) or focus-forming assay (FFA) and plaque-forming unit (PFU) per ml or focus-forming unit (FFU) per ml as inter-changeable terminologies. We have made the changes as suggested by the reviewer. Also, we have indicated aquacide as the overlay medium used in the manuscript; as aquacide is a commercial name for carboxymethylcellulose. We have reworded the sentence in the revised manuscript (Line no. 173, page no. 8).

(7) The pan-flavivirus 4G2 antibody was used in the ADE assay, but the antibody HB112 appears in the experimental results section and figures.

Response: Pan-flavivirus 4G2 antibody is also referred to as HB112 antibody which is the ID of the clone from which it is derived. We have indicated pan-flavivirus 4G2 antibody as HB112 antibody throughout the text.

Response to the Reviewer 1:

Reviewer #1: This manuscript written by Chelluboina et al. describes a basic study on the evaluating ADE activity against dengue virus. Authors compared some protocols of ADE assay using several Fcγ receptor-expressing cell lines (K562, U937 and Vero-CD32a cells). They also compared neutralizing activities between normal Vero and Vero-CD32a cells. They concluded that K562 cells were the most suitable to be used for evaluating ADE activity against dengue virus, and Vero-CD32a cells showed significantly lower neutralizing activity than normal Vero cells when DENV-2 was used as an assay antigen. Possibly the experimental design was elementary, thus the data were not exciting like I have ever seen before. I could not find any unique points which differentiated from other studies. Specific comments are below.

Response: Thank you for the comments. Although several studies have been done using Fc�R-expressing cell lines, here we present an evaluation of two commonly used methods to measure ADE response and compare our results with published literature by evaluating the different Fc�R expressing cell lines (Line no. 566 – 582, page no. 26).

We have added a new Table 3, that clearly demonstrates the sub-neutralizing levels of antibodies leading to antibody enhancement and peak enhancement at different dilutions varied among four serotypes of DENV.

Major comments

(1) Lines 132-135: Information of the Vero-CD32a cells is needed further. Please add a reference describing the origin of Vero-CD32a cells.

Response: Vero cells expressing CD32a receptor (Vero-CD32a cells) were received from Dr. Stephen Whitehead, Laboratory of Viral Diseases, NIH, USA. To generate stable cell lines, Vero cells were transfected with CD32a plasmid, followed by selection in the presence of 50 mg/mL G418. This is an unpublished information. We have added this information in the revised manuscript (Line no. 148-149, page no. 7).

(2) Lines 256-257 and Fig. 2A: Authors should repeat the ADE assay by reducing viral tier in the control.

Response: We have repeated the ADE assay using HB112 mAb by reducing the input DENV-2 virus at MOI=0.05 in Vero and Vero CD32a cell lines. We observed peak-enhancement of DENV-2 infection at 0.1µg/ml HB112 concentration as reported by others (Moi et al, 2011 J Infect Dis. https://academic.oup.com/jid/article/203/10/1405/817308). Please find the image below for your reference.

(3) The ADE pattern of dose response curves in Figs. 3A and 3B did not match the micrographs in Figs. 2A and 2B, although there were some differences in the concentration of HB112. Since the gap will make it difficult for readers to understand the ADE assay, these data should be correlated.

Response: As pointed out correctly, the dose response curves in Fig 3A and 3B look different than the representative image indicated in Fig 2A and 2B due to the differences in the range of concentration of the antibody used in experiments. We have added the new figure panel, 2C of the same concentration as shown in Figures 3A and 3B.

A preliminary experiment of the use of different cell seeding densities showed comparable levels of DENV-2 infection enhancement in U937 and K562 cells (Figure 2B). However, when multiple experiments were performed using the two cell lines, U937 and K562 at 20,000 cells/well seeding density, K562 appears to be superior to U937 cells. This data is already presented in figures 3A and 3B. We have replaced figures 2A and 2B in the revised manuscript.

(4) Figs. 4E and 4F suggested that K562 cells likely showed higher fold enhancement than U937 cells. However, micrographs in Figs. 2A and 2B indicated that U937 cells clearly displayed stronger ADE (fold enhancement) than K562 cells, which suggests that U937 cells may be more sensitive to ADE assay compared with K562 cells. This phenomenon is opposite to Figs. 4E and 4F. What is the cause of this difference? Did the difference between serum and MAb make this? Authors need to consider these points.

Response: We have presented a representative image in Figs. 2A and 2B that is giving contrasting results, with U937 being more sensitive to ADE assay than K562 cells. Sorry about that.

We have revised figures 2A, and 2B for better clarity. Data shown in both Figs.3 and 4 indicates that HB112 (monoclonal antibody, Fig.3) or donor samples (Fig.4) showed a higher degree of fold-enhancement towards DENV-2 and DENV-4 infections when K562 cells were used.

Minor comments

(1) How many cells (cells/well) did you use in the assays of Fig. 2A? (20,000 cells/well)

“CELLS/ML” should be “cells/well” in Fig. 2B

Response: We have used 20,000 cells/well in fig.2A. We have added this information in figure legend. Thank you for pointing out the typo-error in Fig.2B. We have revised figure 2B as suggested.

Response to the Reviewer 2:

Reviewer #2: 1. The manuscript titled "Comparative evaluation of methods for the measurement of antibody-dependent enhancement of dengue virus infection in FcγRIIa expressing cell lines" demonstrate the utilization of different cell lines for ADE analysis. The title itself is little misleading. Actually, the authors have performed similar methodology using different cell lines, so it may be corrected.

Response: We've made changes to the title as suggested.

2. In the manuscript, it is concluded that the infection-ADE assay using K562 cells is the most suitable method to determine the enhancement of infection at sub-neutralizing dilutions, however it is directly corelated with the amount of FcγRIIa expression in the cells, which is quite obvious.

Response: Several Fc�RIIa expressing cell lines such as K562, U937, THP-1, BHK-21, and CV-1 cells have been used to measure the ADE response. Cell lines such as K562, U937, and THP-1 cells naturally express Fc�RIIa receptor alone or in the presence of other receptors such as Fc�RI. While other cell lines such as BHK-21 or CV-1 cells have been genetically modified to express the Fc�RIIa receptors. These modified cell lines resulted in lesser expression of Fc�RIIa as depicted in Figure 1A for Vero-CD32a cells. Simultaneous experiments in two cell lines, U937 and K562 provide direct evidence of the positive correlation of Fc�RIIa expression with enhancement of infection in ADE assay.

We have discussed this point in the revised manuscript (Line no. 567-573, page no. 26).

3. The cross-reactivity should be analyzed with all the four DENV serotypes or at least using VLPs.

Response: We have performed neutralization and ADE assay against all four serotypes and presented the data in Table 3 in the revised manuscript to address the cross-reactivity phenomena in dengue infections.

4. Discussion part of the manuscript is written well, however, Abstract, Introduction, methodology, and result sections need revision in terms of clarity.

Response: We have tried our best to improve the text in the abstract, introduction, methodology, and results sections in the revised manuscript.

---

## [Decision Letter · Decision Letter 1]

14 Mar 2025

PONE-D-24-39295R1Evaluation of methods for the measurement of antibody-dependent enhancement of dengue virus infection using different FcγRIIa expressing cell linesPLOS ONE

Dear Dr. Shrivastava,

Thank you for submitting your manuscript to PLOS ONE. After careful consideration, we feel that it has merit but does not fully meet PLOS ONE’s publication criteria as it currently stands. Therefore, we invite you to submit a revised version of the manuscript that addresses the points raised during the review process. Clarify the results of  ADE experiments in the two cell lines and improve the discussion

We look forward to receiving your revised manuscript.

Kind regards,

Victoria Pando-Robles, Ph.D.

Academic Editor

PLOS ONE

Reviewers' comments:

Reviewer's Responses to Questions

**Comments to the Author**

1. If the authors have adequately addressed your comments raised in a previous round of review and you feel that this manuscript is now acceptable for publication, you may indicate that here to bypass the “Comments to the Author” section, enter your conflict of interest statement in the “Confidential to Editor” section, and submit your "Accept" recommendation.

Reviewer #1: All comments have been addressed

Reviewer #2: All comments have been addressed

2. Is the manuscript technically sound, and do the data support the conclusions?

Reviewer #1: Partly

Reviewer #2: Yes

3. Has the statistical analysis been performed appropriately and rigorously? 

Reviewer #1: I Don't Know

Reviewer #2: Yes

4. Have the authors made all data underlying the findings in their manuscript fully available?

Reviewer #1: No

Reviewer #2: Yes

5. Is the manuscript presented in an intelligible fashion and written in standard English?

Reviewer #1: Yes

Reviewer #2: Yes

6. Review Comments to the Author

Reviewer #1: (1) Lines 148-149: For the reproductive experiment by other researches in the future, plasmid information (CD32a plasmid) should be disclosed.

(2) Figure 2A: Figure 2A was not replaced successfully in the revised manuscript. “No Ab control” has been still saturated in Vero-CD32a and Vero cells. If the saturated condition was relieved, you are able to state the distinction clearly (lines 278-279). As far as I observed your new data in the response, Vero-CD32a did not show any ADE activities. Authors need to discuss a relationship between ADE and Vero-CD32a. Furthermore, there is a mismatch regarding MOI between their response (MOI=0.05) and manuscript text (1 MOI [line 278]). Authors need to revise carefully the figure legend, too.

(3) (4) Figures 2A and 2B were not replaced in the revised manuscript.

Reviewer #2: Authors have addressed all the queries and revised the manuscript accordingly. Therefore, it may be accepted for the publication.

7. PLOS authors have the option to publish the peer review history of their article (what does this mean? ). If published, this will include your full peer review and any attached files.

**Do you want your identity to be public for this peer review?** For information about this choice, including consent withdrawal, please see our Privacy Policy .

Reviewer #1: No

Reviewer #2: No

---

## [Author Response · Author response to Decision Letter 2]

20 Mar 2025

Response to the Academic editor:

1. Clarify the results of ADE experiments in the two cell lines and improve the discussion

Response: We have improved the discussion as suggested in the revised manuscript, line no. 648-663, page no. 28.

Response to the Reviewer 1:

(1) Lines 148-149: For the reproductive experiment by other researches in the future, plasmid information (CD32a plasmid) should be disclosed.

Response: Vero cells expressing CD32a receptors (Vero-CD32a cells) were received from the Laboratory of Viral Diseases, at the NIH. A sequence encoding the human FcγR CD32a receptor was cloned into the mammalian expression vector pT-Rex DEST30 (Invitrogen). To generate stable cell lines, Vero cells were transfected with CD32a plasmid, followed by selection in the presence of 50 mg/mL G418. This is an unpublished information. As of now, information on these cells is mentioned in the thesis published online which has been cited in the manuscript (Reference no 28).

(2) Figure 2A: Figure 2A was not replaced successfully in the revised manuscript. “No Ab control” has been still saturated in Vero-CD32a and Vero cells. If the saturated condition was relieved, you are able to state the distinction clearly (lines 278-279). As far as I observed your new data in the response, Vero-CD32a did not show any ADE activities. Authors need to discuss a relationship between ADE and Vero-CD32a. Furthermore, there is a mismatch regarding MOI between their response (MOI=0.05) and manuscript text (1 MOI [line 278]). Authors need to revise carefully the figure legend, too.

Response: We have revised figure 2A with lower MOI infection in both Vero and Vero-CD32a cells. As pointed out correctly by the reviewer, we did not observe any ADE activity in Vero-CD32a cells even at lower MOI infection. We have revised the statement on ADE response in Vero CD32a and Vero cells at both MOI 1 and MOI 0.05 for a better clarity in revised manuscript. (Line no. 283-289, page no. 12). We have revised figure legend too.

(3) (4) Figures 2A and 2B were not replaced in the revised manuscript.

Response: We have modified Figure 2A. Also, Figure 2B was modified and separate panel as Figure 2C was added after the first revision. We have re-submitted all revised figures with the manuscript.

Reviewer #2: Authors have addressed all the queries and revised the manuscript accordingly. Therefore, it may be accepted for the publication.

Response: Thank you for your kind response.

---

## [Decision Letter · Decision Letter 2]

21 Apr 2025

PONE-D-24-39295R2Evaluation of methods for the measurement of antibody-dependent enhancement of dengue virus infection using different FcγRIIa expressing cell linesPLOS ONE

Dear Dr. Shrivastava,

Thank you for submitting your manuscript to PLOS ONE. After careful consideration, we feel that it has merit but does not fully meet PLOS ONE’s publication criteria as it currently stands. Therefore, we invite you to submit a revised version of the manuscript that addresses the points raised during the review process.

It remains unclear whether Vero-CD32 a cells exhibit ADE for dengue virus. Please clarify this point and provide an explanation for the results shown in Figures 7 and 8.

Please submit your revised manuscript by Jun 05 2025 11:59PM.  If you will need more time than this to complete your revisions, please reply to this message or contact the journal office at plosone@plos.org . Please include the following items when submitting your revised manuscript:

We look forward to receiving your revised manuscript.

Kind regards,

Victoria Pando-Robles, Ph.D.

Academic Editor

PLOS ONE

Reviewers' comments:

Reviewer's Responses to Questions

**Comments to the Author**

1. If the authors have adequately addressed your comments raised in a previous round of review and you feel that this manuscript is now acceptable for publication, you may indicate that here to bypass the “Comments to the Author” section, enter your conflict of interest statement in the “Confidential to Editor” section, and submit your "Accept" recommendation.

Reviewer #1: All comments have been addressed

Reviewer #3: (No Response)

2. Is the manuscript technically sound, and do the data support the conclusions?

Reviewer #1: Partly

Reviewer #3: (No Response)

3. Has the statistical analysis been performed appropriately and rigorously? 

Reviewer #1: I Don't Know

Reviewer #3: (No Response)

4. Have the authors made all data underlying the findings in their manuscript fully available?

Reviewer #1: No

Reviewer #3: (No Response)

5. Is the manuscript presented in an intelligible fashion and written in standard English?

Reviewer #1: Yes

Reviewer #3: (No Response)

6. Review Comments to the Author

Reviewer #1: Authors have responded to most of the reviewer’s comments. However, still I do not understand one point. They need to indicate clearly whether Vero-CD32a cells display ADE or not. Their present statement will make readers confused. Several sentences in the manuscript are contradictory (see below). Especially, Figs. 8C and 8D simply showed result of FRNT, thus the authors’ statement regarding infection-enhancing activity was misleading (Lines 465-466).

Lines 283-284: “no distinct enhancement pattern was observed at any HB112 concentrations in Vero-CD32a cells to No Ab control wells”.

Lines 465-466: “This data suggests that low infection-enhancing activity was noted in Vero-CD32a cells.”

Lines 627-629: “It was observed that follow-up samples collected after 10 years post-vaccination showed enhancement of infection in Vero-CD32a cells when compared to sera collected at day 208 post-vaccination”.

Reviewer #3: This study evaluates methodologies for measuring ADE of DENV infection using different FcyRIIa-expressing cell lines (U937, K562, and Vero-CD32a). The authors compared ADE responses in plasma from healthy donors (n=12) and sera from secondary dengue patients (n=12). Results demonstrated that K562 cells were superior to U937 and Vero-CD32a for detecting ADE, particularly for DENV-4. Secondary dengue samples exhibited high neutralizing activity in Vero cells but low enhancement in Vero-CD32a. The study highlights the role of sub-neutralizing antibody concentrations in ADE and proposes K562 cells as a reliable tool for ADE assessment.

Comments:

1. The manuscript reports that Vero-CD32a cells showed no distinct ADE activity, even at lower MOIs (Figure 2A), yet later states that some secondary dengue samples exhibited infection-enhancing activity in Vero-CD32a cells (Table 2). This contradiction is not addressed, leading to confusion about the utility of Vero-CD32a cells for ADE measurement.

2. The manuscript concludes that K562 cells are superior for ADE assays due to higher FcγRIIa expression and lack of viral replication in the absence of antibodies. However, it does not explore the mechanistic reasons for this superiority, such as differences in FcγRIIa signaling, internalization efficiency, or cell-specific responses to DENV. The claim that K562 cells only express FcγRII is not supported by a reference or experimental validation.

3. The manuscript reports significantly lower neutralizing antibody titers in Vero-CD32a compared to Vero cells for DENV-2 (Line 430) but not for DENV-4 (Line 435). This contradicts prior studies that found lower neutralizing titers in FcγRIIa-expressing cells for all serotypes due to ADE. The authors do not explain why DENV-4 behaves differently or reconcile their findings with existing literature.

4. The optimization of the ADE assay in Vero-CD32a cells is inadequate. The manuscript notes that "No Ab control wells were fully saturated with infection at 1 MOI" (Line 280), and even at a lower MOI (0.05), no ADE was observed (Line 283). This suggests that the assay conditions were not optimized for Vero-CD32a, potentially skewing the comparison with K562 and U937 cells. Additionally, the cut-off for fold-enhancement (1.8 for Vero-CD32a, 10 for K562/U937) differs between assays without justification.

5. The manuscript briefly mentions the limitation of not knowing the infecting serotypes for secondary dengue patients (Lines 653-656) but does not discuss the implications of this gap. For example, cross-reactive antibodies could skew ADE results, particularly in hyperendemic areas. Additionally, the study does not address the potential impact of FcγRIIa polymorphisms (e.g., R131H), which are known to influence ADE.

6. The presentation of results, particularly in Figures 7 and 8, is confusing. For example, Figure 7A shows a significant difference in FRNT50 titers for DENV-2 between Vero and Vero-CD32a cells, but the text does not explain why only 33% of samples showed a reduction (Line 426). Figure 8C-D reports comparable titers for DENV-2 but a significant difference for DENV-4, without clear justification. Additionally, the use of geometric mean titers obscures individual sample variability.

7. PLOS authors have the option to publish the peer review history of their article (what does this mean? ). If published, this will include your full peer review and any attached files.

**Do you want your identity to be public for this peer review?** For information about this choice, including consent withdrawal, please see our Privacy Policy .

Reviewer #1: No

Reviewer #3: No

---

## [Author Response · Author response to Decision Letter 3]

2 Jul 2025

Dear Editor and Reviewers,

Thank you for your valuable feedback on our manuscript titled "Comparative evaluation of methods for the measurement of antibody-dependent enhancement of dengue virus infection in FcγRIIa expressing cell lines" (Manuscript ID: PONE-D-24-39295 PLOS ONE). We appreciate the time and effort you have invested in reviewing our work. We have carefully considered all the comments and have made significant revisions to address them.

Response to the Academic editor:

1. It remains unclear whether Vero-CD32a cells exhibit ADE for dengue virus. Please clarify this point and provide an explanation for the results shown in Figures 7 and 8.

Response:

In the current manuscript, we attempted to evaluate different cell lines and different test formats for their ability to exhibit ADE using the same set of samples. In the ADE-infection assay, initially only two cell lines, K562 and U937, were compared. We have performed ADE-infection assay using Vero-CD32a, and the data were compared against the other two cell lines (revised Figure 4). Our data demonstrate that the enhancing activity was seen with K562 and U937 cells, whereas Vero-CD32a cells were unable to exhibit ADE (revised Figure 4).

When FRNT was performed simultaneously in Vero and Vero-CD32a cells, individual samples did show enhancing activity in Vero-CD32a cells, but the magnitude of fold enhancement was minimal, up to 2-fold in Vero-CD32a cells (Figures 7 and 8). When we did paired analyses of GMT titres employing Wilcoxon signed rank test in blood donor samples, no significant differences were observed (revised Figure 7). Similarly, comparable DENV-2 titres and significantly lower DENV-4 titres were obtained in Vero-CD32a than in Vero cells when secondary dengue samples were analysed (revised Figure 8).

Thus, the infection-ADE assay is a suitable method for ADE assessment. Comparison of FRNT50 titres in Vero and Vero-CD32a cells did not show distinct enhancing activity, probably due to the samples having high neutralizing activity. We have revised the manuscript to bring clarity in the manuscript.

Response to the Reviewer 1:

Reviewer #1: Authors have responded to most of the reviewer’s comments. However, still I do not understand one point. They need to indicate clearly whether Vero-CD32a cells display ADE or not. Their present statement will make readers confused. Several sentences in the manuscript are contradictory (see below). Especially, Figs. 8C and 8D simply showed result of FRNT, thus the authors’ statement regarding infection-enhancing activity was misleading (Lines 465-466).

Lines 283-284: “no distinct enhancement pattern was observed at any HB112 concentrations in Vero-CD32a cells to No Ab control wells”.

Lines 465-466: “This data suggests that low infection-enhancing activity was noted in Vero-CD32a cells.”

Lines 627-629: “It was observed that follow-up samples collected after 10 years post-vaccination showed enhancement of infection in Vero-CD32a cells when compared to sera collected at day 208 post-vaccination”.

Response: We have revised the contradictory statements. Individual samples did show enhancing activity in FRNT method, however, when a non-parametric Wilcoxon signed rank test was performed, titres were found comparable in both Vero and Vero-CD32a cells against DENV-2 (Fig. 8C), but not with DENV-4 viruses (Fig. 8D). Enhancing activity is influenced by DENV serotypes and we have discussed this point in the revised manuscript (line no. 628-637, page no. 28). We have revised the line no. 283-284, 465-466 and line no. 627-629 for better clarity.

Response to the Reviewer 3:

1. The manuscript reports that Vero-CD32a cells showed no distinct ADE activity, even at lower MOIs (Figure 2A), yet later states that some secondary dengue samples exhibited infection-enhancing activity in Vero-CD32a cells (Table 2). This contradiction is not addressed, leading to confusion about the utility of Vero-CD32a cells for ADE measurement.

Response: As indicated in Figure 2A, Vero-CD32a did not demonstrate any enhancing activity at any concentrations of HB112 (4G2) antibody, even at lower MOI of 0.05. Moi et al (2013) demonstrated that monoclonal antibody concentration but not the challenge virus titres influence the ADE activity of either monoclonal antibodies or dengue patients’ serum samples.

In fact, we followed Moi et al (2013) methodology to measure the in-vitro ADE activity of human serum samples by conventional plaque assay using Fc�R-expressing cells. We used FRNT data to re-analyze in terms of virus titres at different dilutions of samples and compared the titres in Vero and Vero-CD32a cells, which led to contradictory results as shown in Table 2.

We have removed Table 2 from the revised manuscript. For ADE measurement, only two methods were evaluated and compared (1) ADE-infection assay from the culture supernatants, and (2) simultaneous performance of FRNT in Vero and Vero-CD32a cells throughout the manuscript, and results were discussed.

Secondary dengue samples showing infection-enhancement activities to a lower extent using Vero-CD32a cells were discussed in the revised manuscript (line no. 628-637, page no. 28).

2. The manuscript concludes that K562 cells are superior for ADE assays due to higher FcγRIIa expression and lack of viral replication in the absence of antibodies. However, it does not explore the mechanistic reasons for this superiority, such as differences in FcγRIIa signaling, internalization efficiency, or cell-specific responses to DENV. The claim that K562 cells only express FcγRII is not supported by a reference or experimental validation.

Response: K562 cells predominantly express Fc�RIIa receptor with an activating cytoplasmic ITAM motif. Boonnak et al (2013) demonstrated that ADE of dengue virus infection was significantly abrogated in K562 cells when either monoclonal antibodies blocking the Fc�RIIa surface expression or siRNA treatment to knockdown Fc�RIIa expression were used. We have provided this information in the revised manuscript (line no. 553-562, page no. 25).

3. The manuscript reports significantly lower neutralizing antibody titers in Vero-CD32a compared to Vero cells for DENV-2 (Line 430) but not for DENV-4 (Line 435). This contradicts prior studies that found lower neutralizing titers in FcγRIIa-expressing cells for all serotypes due to ADE. The authors do not explain why DENV-4 behaves differently or reconcile their findings with existing literature.

Response: We have explained the contradictory results obtained in our study with prior research articles (Line no. 616-637, Page no. 28).

4. The optimization of the ADE assay in Vero-CD32a cells is inadequate. The manuscript notes that "No Ab control wells were fully saturated with infection at 1 MOI" (Line 280), and even at a lower MOI (0.05), no ADE was observed (Line 283). This suggests that the assay conditions were not optimized for Vero-CD32a, potentially skewing the comparison with K562 and U937 cells. Additionally, the cut-off for fold-enhancement (1.8 for Vero-CD32a, 10 for K562/U937) differs between assays without justification.

Response: Since, DENV infection enhancement activity could be influenced by input virus MOI, antibody concentration, and virus serotype, we optimized the ADE assay using DENV-2 infection at different MOI and HB112 (4G2) antibody concentrations in K562, U937, and Vero-CD32a cell lines. Distinct enhancement of DENV-2 infection was seen in K562 and U937 cells but not in Vero-CD32a cells (Revised Figure 4).

As described earlier, for the evaluation of virus titres at different dilutions using FRNT data, cut-off value of 1.8 for fold-enhancement was used following Moi et al (2013) paper. We have removed this analysis from the revised manuscript for better clarity.

5. The manuscript briefly mentions the limitation of not knowing the infecting serotypes for secondary dengue patients (Lines 653-656) but does not discuss the implications of this gap. For example, cross-reactive antibodies could skew ADE results, particularly in hyperendemic areas. Additionally, the study does not address the potential impact of FcγRIIa polymorphisms (e.g., R131H), which are known to influence ADE.

Response: We have discussed these points in the revised manuscript (line no. 676-696, line no. 30).

6. The presentation of results, particularly in Figures 7 and 8, is confusing. For example, Figure 7A shows a significant difference in FRNT50 titers for DENV-2 between Vero and Vero-CD32a cells, but the text does not explain why only 33% of samples showed a reduction (Line 426). Figure 8C-D reports comparable titers for DENV-2 but a significant difference for DENV-4, without clear justification. Additionally, the use of geometric mean titers obscures individual sample variability.

Response: We have revised Figures 7 and 8, a non-parametric Wilcoxon signed rank test was performed to obtain statistical value, donor samples did not exhibit enhancing activity in Vero-CD32a cells (revised Figure 7).

There is a significant difference in titres against DENV-4 in secondary dengue samples, but not with DENV-2 serotype. There are multiple factors involved like sample type, previous primary or secondary DENV immunity, quantity of cross-reactive neutralizing antibodies, and serotype-specific differences in neutralizing antibody response that could lead to differences in serotype-specific titres. We have provided justification in the revised manuscript (line no. 618-637, page no. 28).

---

## [Decision Letter · Decision Letter 3]

14 Aug 2025

Evaluation of methods for the measurement of antibody-dependent enhancement of dengue virus infection using different FcγRIIa expressing cell lines

PONE-D-24-39295R3

Dear Dr. Shrivastava,

We’re pleased to inform you that your manuscript has been judged scientifically suitable for publication and will be formally accepted for publication once it meets all outstanding technical requirements.

Kind regards,

Victoria Pando-Robles, Ph.D.

Academic Editor

PLOS ONE

Additional Editor Comments (optional):

Reviewers' comments:

Reviewer's Responses to Questions

**Comments to the Author**

1. If the authors have adequately addressed your comments raised in a previous round of review and you feel that this manuscript is now acceptable for publication, you may indicate that here to bypass the “Comments to the Author” section, enter your conflict of interest statement in the “Confidential to Editor” section, and submit your "Accept" recommendation.

Reviewer #1: All comments have been addressed

2. Is the manuscript technically sound, and do the data support the conclusions?

Reviewer #1: Yes

3. Has the statistical analysis been performed appropriately and rigorously? 

Reviewer #1: Yes

4. Have the authors made all data underlying the findings in their manuscript fully available?

Reviewer #1: Yes

5. Is the manuscript presented in an intelligible fashion and written in standard English?

Reviewer #1: Yes

6. Review Comments to the Author

Reviewer #1: (No Response)

7. PLOS authors have the option to publish the peer review history of their article (what does this mean? ). If published, this will include your full peer review and any attached files.

**Do you want your identity to be public for this peer review?** For information about this choice, including consent withdrawal, please see our Privacy Policy .

Reviewer #1: No

---

## [Editor Report · Acceptance letter]

PONE-D-24-39295R3

PLOS ONE

Dear Dr. Shrivastava,

I'm pleased to inform you that your manuscript has been deemed suitable for publication in PLOS ONE. Congratulations! Your manuscript is now being handed over to our production team.

Kind regards,

on behalf of

Dr. Victoria Pando-Robles

Academic Editor

PLOS ONE